# The ATM and ATR kinases regulate centrosome clustering and tumor recurrence by targeting KIFC1 phosphorylation

Guangjian Fan [1,8], Lianhui Sun[1,8], Ling Meng[2,8], Chen Hu[1], Xing Wang[1], Zhan Shi[1], Congli Hu[1], Yang Han[1], Qingqing Yang[1], Liu Cao[3], Xiaohong Zhang[4], Yan Zhang[5], Xianmin Song[5], Shujie Xia[6], Baokun He[7], Shengping Zhang [1✉] & Chuangui Wang [1✉]

Drug resistance and tumor recurrence are major challenges in cancer treatment. Cancer cells often display centrosome amplification. To maintain survival, cancer cells achieve bipolar division by clustering supernumerary centrosomes. Targeting centrosome clustering is therefore considered a promising therapeutic strategy. However, the regulatory mechanisms of centrosome clustering remain unclear. Here we report that KIFC1, a centrosome clustering regulator, is positively associated with tumor recurrence. Under DNA damaging treatments, the ATM and ATR kinases phosphorylate KIFC1 at Ser26 to selectively maintain the survival of cancer cells with amplified centrosomes via centrosome clustering, leading to drug resistance and tumor recurrence. Inhibition of KIFC1 phosphorylation represses centrosome clustering and tumor recurrence. This study identified KIFC1 as a prognostic tumor recurrence marker, and revealed that tumors can acquire therapeutic resistance and recurrence via triggering centrosome clustering under DNA damage stresses, suggesting that blocking KIFC1 phosphorylation may open a new vista for cancer therapy.

[1] Translational Medicine Center, Shanghai General Hospital, Shanghai Jiao Tong University School of Medicine, 201620 Shanghai, China. [2] Department of Pulmonary and Critical Care Medicine, The Second Affiliated Hospital of Shandong First Medical University, 271000 Shandong, China. [3] Key Laboratory of Medical Cell Biology, College of Translational Medicine, China Medical University, 110000 Shenyang, China. [4] Department of Oncology, Karmanos Cancer Institute, Wayne State University School of Medicine, 4100 John R., Detroit, MI 48201, USA. [5] Department of Hematology, Shanghai General Hospital, Shanghai Jiao Tong University School of Medicine, 201620 Shanghai, China. [6] Department of Urology, Shanghai General Hospital, Shanghai Jiao Tong University School of Medicine; Institute of Urology, Shanghai Jiao Tong University, 200080 Shanghai, China. [7] Department of Gastroenterology, Shanghai General Hospital, Shanghai Jiao Tong University School of Medicine, 201620 Shanghai, China. [8]These authors contributed equally: Guangjian Fan, Lianhui Sun, Ling Meng. ✉email: spzhang2019@126.com; cgwang2016@sjtu.edu.cn

Chromosomal instability (CIN) is widely recognized as a hallmark of many tumors[1–5]. CIN contributes to tumor heterogeneity and is positively associated with increased radiation-resistance or drug-resistance, poor patient prognosis, and high risk of tumor recurrence[6–8]. Supernumerary centrosomes (generating more than two centrosomes) are common in human cancers and are correlated with CIN. Cancer cells with supernumerary centrosomes can either die due to high levels of aneuploidy generated by multipolar mitosis or divide and produce viable progeny by achieving a pseudo-bipolar structure via clustering their centrosomes into two functional poles, a process called centrosome clustering[9–11]. Cancer cells undergoing centrosome clustering have a prolonged time to form a pseudo-bipolar spindle in which single kinetochores often attach to microtubules emanating from different poles, giving rise to lagging chromosomes during anaphase[9,10]. Increasing evidence has shown that centrosome clustering is an essential mechanism for CIN, and targeting centrosome clustering is considered a promising means for therapeutic intervention[10,12]. However, the regulatory mechanism of centrosome clustering and its function in cancer therapy is largely unknown. In addition, increasing evidence has demonstrated that DNA damage promotes centrosome amplification[13–15], and the cancer cells undergoing centrosome clustering often acquire invasive properties[2]. Finding the driving mechanism of centrosome clustering is therefore urgently needed to guide precision cancer therapy.

The kinesin-like protein KIFC1, a nonessential minus end-directed motor of the kinesin-14 family[10,12,16,17], has recently emerged as a crucial player in the bi-focal clustering of supernumerary centrosomes in human cancer cells during mitosis[10,18,19]. Intriguingly, depletion of KIFC1 induces a dramatic increase in multipolar anaphases and triggers cancer cell death due to catastrophic multipolar division, whereas knockdown of KIFC1 has little effect on cell division in cells with two centrosomes[10]. Clinical specimens show that KIFC1 is highly expressed in ovarian adenocarcinomas, hepatocellular carcinoma, and breast cancer[20–22]. Thus, inhibition of KIFC1 is a promising strategy to prevent CIN and cellular invasion in cancer therapy. Unfortunately, the available KIFC1 inhibitors either show a lack of potency and specificity or have unfavorable toxicity effects[18,23,24]. A comprehensive study of regulatory mechanisms controlling KIFC1 expression and function may therefore provide new insights into the prevention of tumor therapy resistance, recurrence, and metastasis.

In this study, we showed that KIFC1 is a potential marker for clinical cancer recurrence. DNA damage-inducing therapies activated DNA damage response kinases ATM and ATR, which phosphorylated of KIFC1 at Ser26. KIFC1 was stabilized upon phosphorylation and thus promoted centrosome clustering, CIN, and tumor recurrence both in vivo and in vitro. Blocking of KIFC1 phosphorylation markedly prevented the DNA damage-induced CIN and tumor recurrence. These results provide new insight into mechanisms regarding how DNA damaging therapies always lead to CIN and therapeutic resistance, suggesting that targeting KIFC1 phosphorylation may provide new opportunities for reducing tumor metastasis and recurrence.

## Results

**KIFC1 predicts human tumor recurrence.** Increasing evidence has indicated that centrosome clustering is an essential source of CIN[10,12] and CIN is closely associated with patient prognosis and tumor recurrence[6–8], so we assessed the prognostic value of KIFC1 using breast ($n = 140$) and colorectal cancer ($n = 83$) tissue microarrays. We stained the tumor tissue microarrays using anti-KIFC1 antibody and classified the specimens into low, medium, and high KIFC1 expression groups according to the immunohistochemical (IHC) scores. The results showed that KIFC1 expression was positively correlated with human breast tumor recurrence, but not with age (Fig. 1a–c). In the tumor recurrence group, about 70% of the breast cancer patients showed a high expression of KIFC1 (IHC score ≥8) (Fig. 1b). The survival of breast cancer patients with a high KIFC1 protein expression was significantly lower than in patients with low KIFC1 expression (Fig. 1d). Similarly, the KIFC1 protein level was positively correlated with colorectal tumor recurrence, but not with age and gender (Fig. 1e–g). The survival of colorectal cancer patients with a high KIFC1 expression was also significantly decreased (Fig. 1h). In addition, the KIFC1 protein level was higher in colorectal tumor tissues than in paired peri-tumor specimens (Fig. S1a, b). These results suggest that high expression of KIFC1 is an adverse prognostic biomarker for tumor recurrence of breast and colorectal cancers.

**DNA-damaging treatments induce the centrosome clustering protein KIFC1.** Tumor recurrence is often related to drug resistance during cancer therapy. We therefore selected four patient-derived xenograft (PDX) models of breast cancer (BRPF212, BRPF280, BRPF232, and BRPF008) (Fig. S2) to analyze the changes in KIFC1 protein levels after treatment with conventional chemotherapeutic drugs including cisplatin and etoposide. The results show that both cisplatin and etoposide treatments significantly enhanced KIFC1 expression in PDX tissues (Fig. 2a), indicating that DNA-damaging therapies promote KIFC1 protein accumulation in vivo.

To confirm the above results, we next examined the effects of various DNA-damaging treatments on KIFC1 expression using several cancer cell lines representing different types of human cancers. The result showed that X-ray irradiation (IR, inducing DNA double-strand breaks[25]), ultraviolet (UV, inducing DNA single-strand breaks[25]), and nine DNA-damaging drugs significantly increased KIFC1 and γH2AX (a DNA damage marker[26]) expressions in human breast cancer MDA-MB-231 cells (Fig. 2b). Moreover, etoposide treatment promoted the KIFC1 expression in six different types of tumor cells (Fig. 2c), and etoposide, cisplatin, and IR treatments markedly enhanced KIFC1 staining in MDA-MB-231 xenograft tumors of nude mice (Fig. 2d). Collectively, these results reveal that DNA-damaging therapies promote KIFC1 accumulation both in vitro and in vivo.

**DNA-damaging treatments induce KIFC1-dependent centrosome clustering.** Previous studies revealed that DNA damage increases centrosome amplification (CA)[13–15], and centrosome clustering is vital for the survival of cancer cells containing supernumerary centrosomes[5]. Because the above data showed that DNA damage treatments caused a significant increase of KIFC1 expression, we next assessed the effects of DNA-damaging treatments in regulating centrosome clustering. Firstly, we confirmed that the percentage of centrosome amplification (>2 centrosomes per cell) increased after DNA damage treatment (36% of CA at 0 h, 41% of CA at 15 h, and 50% of CA at 48 h) in MDA-MB-231 cells (Fig. S3a). Notably, the ratio of centrosome clustering to non-efficient centrosome clustering was increased significantly at 15 h and 48 h after etoposide treatment in mitotic cells (Fig. S3b). To exclude the influence of apoptotic cells induced by DNA-damaging agents at 48 h, we selected 15 h after etoposide treatment to precisely and conveniently examine the frequency of centrosome clustering. The results showed that IR and etoposide treatments significantly increased the frequency of mitotic cells with centrosome clustering, whereas it decreased the

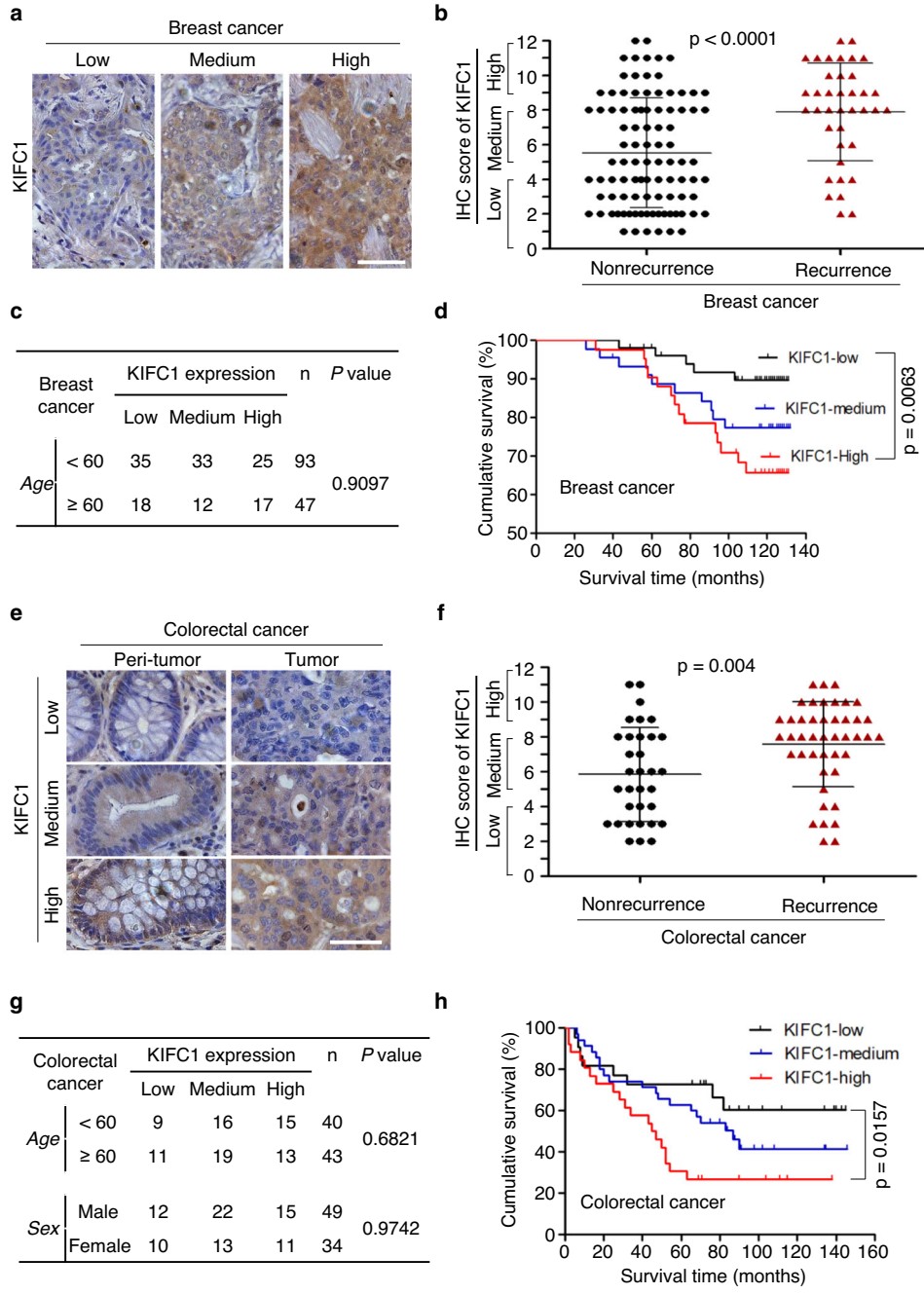

**Fig. 1 KIFC1 predicts human tumor recurrence.** The correlation of KIFC1 protein expression and tumor recurrence or poor prognosis in breast cancer (*n* = 140, **a**–**d**) or colorectal cancer (*n* = 83, **e**–**h**) tissue chips. **a, b, e, f** Data are representative of the KIFC1 stained tumor or adjacent non-tumor tissues (**a, e**) (scale bar, 50 μm) with quantitative analysis of clinical samples of tumor recurrence or non-recurrence (**b, f**). KIFC1 expression was classified as high, medium, or low according to the staining signals. Two-tailed *t* test *p* values: *p* = <0.00001 (**b**). **c, d, g, h** The tables show the relative levels of KIFC1 expression with patient age or gender in breast cancers (**c**) or colorectal cancers (**g**). Kaplan–Meier plot of overall survival of patients with breast cancer (**d**) or colorectal cancer (**h**) stratified by the KIFC1 expression level. A log-rank test was used for statistical analysis. All data presented in this figure show mean values ± SD. Statistical significance was determined by Two-tailed *t* test. Source data are provided as a Source Data file.

occurrence of non-efficient centrosome clustering (multipolar anaphases, as indicated by centrin staining) in multiple types of cancer cell lines (Fig. 3a, b). Similar results were observed in cells treated with other DNA-damaging treatments (including cisplatin, oxaliplatin, mitomycin C, estramustine, epirubicin, gemcitabine, bleomycin, and CTX) (Fig. 3c). Moreover, we observed that knockdown of KIFC1 decreased etoposide-induced enhancement of centrosome clustering coupled with an increase of multipolar centrosomes in etoposide-treated cells (Fig. 3d), suggesting that KIFC1 was required for etoposide-induced centrosome clustering. Furthermore, in vivo studies using PDX tissues (Fig. 3e) and the MDA-MB-231 xenograft tumors (Fig. 3f) in nude mice also confirmed that DNA damaging treatments induced marked enhancement of centrosome clustering coupled with a decrease of non-efficient clustering. Collectively, these results suggest that DNA-damaging treatments cause enhanced KIFC1 expression, leading to KIFC1-dependent centrosome clustering in tumor cells.

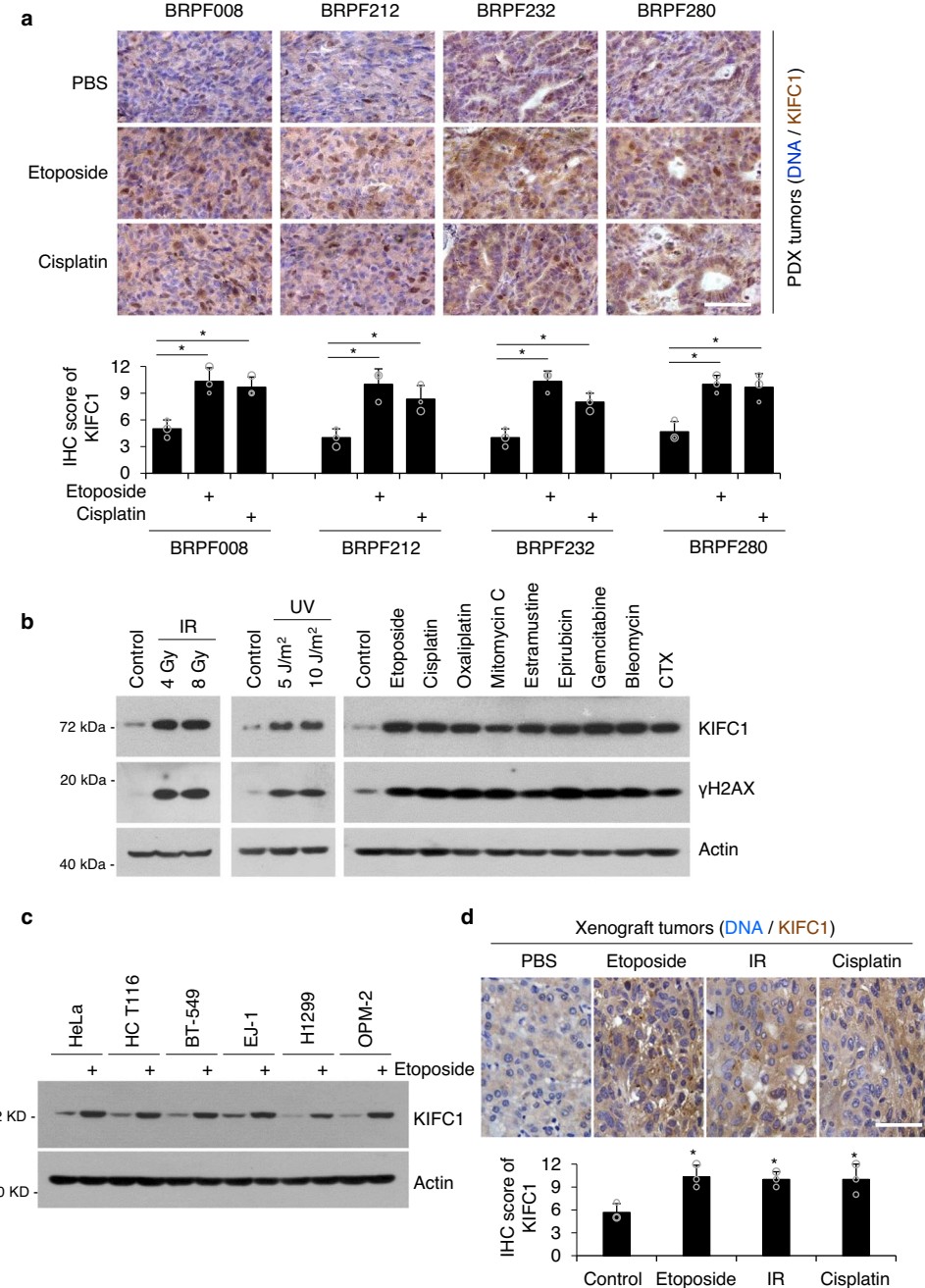

**Fig. 2 DNA-damaging treatments induce the centrosome clustering protein KIFC1. a** KIFC1 protein expression in PDX models after treatment with PBS, etoposide, or cisplatin. Data are representative of stained tumor tissues (scale bar, 50 μm) with quantitative analysis. Two-tailed $t$ test $p$ values: $p =$ 0.04721 (BRPF008, Eto), 0.0339 (008, Cis), 0.0350 (212, Eto), 0.0390 (212, Cis), 0.0341 (232, Eto), 0.0202 (232, Cis), 0.0263 (280, Eto), and 0.0130 (280, Cis). (**b, c**) MDA-MB-231 cells (**b**) or indicated cells (**c**) were treated with irradiation (IR), ultraviolet (UV) light, etoposide (10 μM), cisplatin (10 μM), oxaliplatin (40 μM), mitomycin C (2 μM), estramustine (20 μM), epirubicin (0.5 μM), gemcitabine (4 μM), bleomycin (10 μM), or cyclophosphamide (CTX, 10 mM) for 15 h. Cell lysates were immunoblotted with antibodies against KIFC1, γH2AX, or β-actin (as the internal standard). **d** MDA-MB-231 xenograft tumors were treated with etoposide, IR, or cisplatin as described in the Methods section. Data are representative of stained tumor tissues (scale bar, 50 μm) with quantitative analysis. Three tumors were included in each group. Two-tailed $t$ test $p$ values: $p = 0.0424$ (Eto), 0.0345 (IR), and 0.0415 (Cis). Data represent the mean ± SD of three independent experiments (**a, d**). *$p < 0.05$, **$p < 0.01$, Source data are provided as a Source Data file.

## ATM and ATR kinases phosphorylate KIFC1-S26 during DNA-damage conditions.

Next, we investigated the mechanism responsible for KIFC1 accumulation after DNA damage. The results showed that etoposide markedly delayed KIFC1 protein degradation (Fig. 4a) without altering its transcription (Fig. 4b). In addition, both etoposide and cisplatin decreased KIFC1 ubi-quitination (Fig. 4c), indicating that DNA damaging agents led to stabilization of KIFC1 protein. The central components of the DNA repair pathway are ataxia telangiectasia mutated (ATM) and ataxia telangiectasia mutated and Rad3-related (ATR) kina-ses[27–29]. ATM and ATR activated by DNA double-strand or single-strand breaks[25] lead to phosphorylation of several key players in DNA damage response, such as CHK1 and CHK2 kinases[26,30]. To investigate whether these kinases regulated the

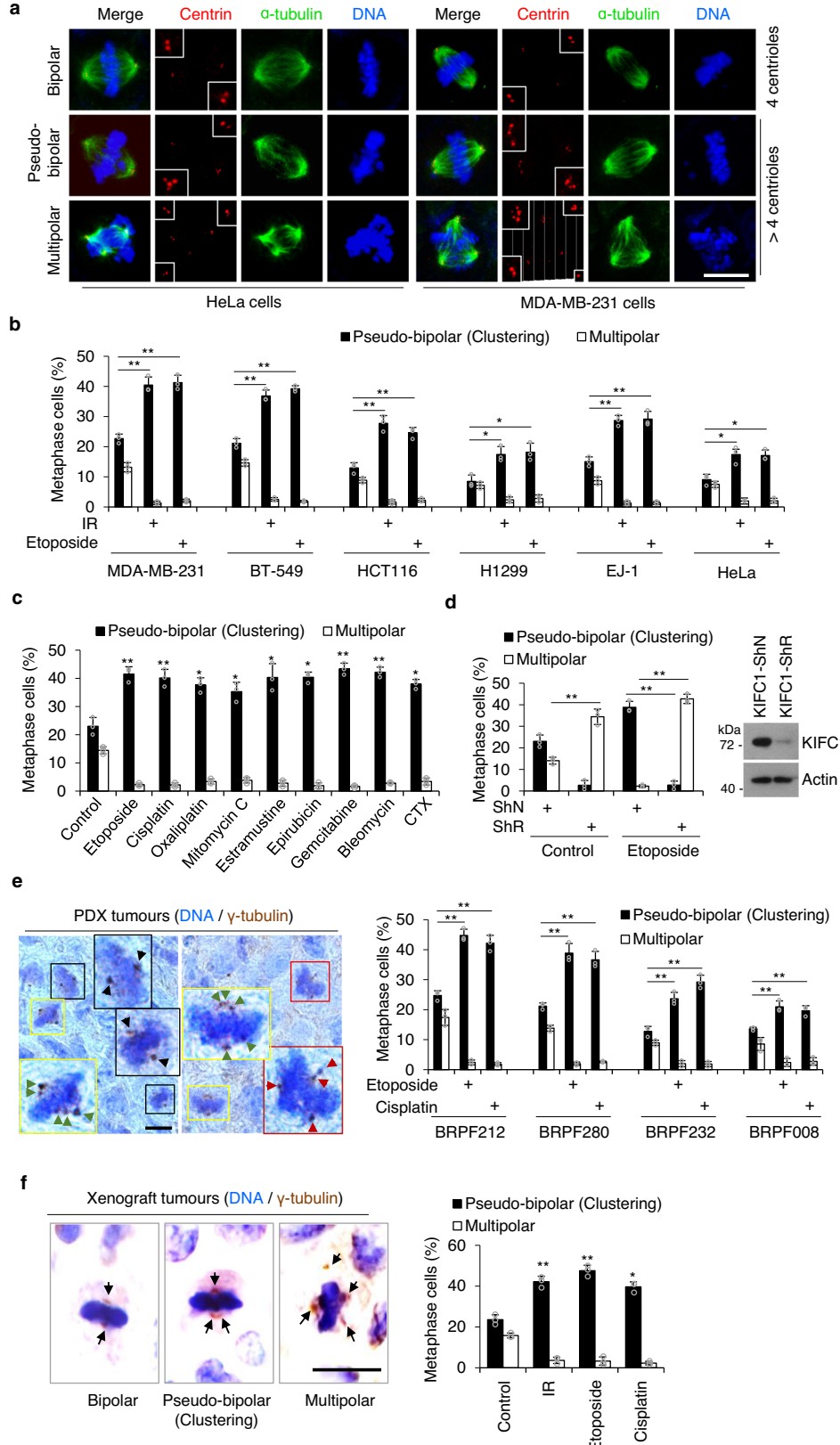

stability of the KIFC1 protein, we used AZD1390 (an ATM inhibitor)[31], AZD6738 (an ATR inhibitor)[32], VE-822 (an inhibitor of ATM and ATR at a concentration of 5 μM)[33–35], MK-8776 (a CHK1 inhibitor), and C3742 (a CHK2 inhibitor). The results showed that etoposide-induced KIFC1 upregulation was markedly inhibited by VE-822, AZD1390, and AZD6738, but not by MK-8776, and C3742 (Fig. 4d). Moreover, knockdown of ATM and ATR markedly decreased etoposide-induced KIFC1

**Fig. 3 DNA-damaging treatments induce KIFC1-dependent centrosome clustering. a** Representative images showing bipolar (4 normal centrioles), pseudo-bipolar (centrosome clustering, >4 centrioles), and multipolar mitosis (non-efficient centrosome clustering, >4 centrioles) in HeLa and MDA-MB-231 cells. Spindle poles, centrioles, and DNA were co-stained with α-tubulin, centrin, and 4′,6-diamidino-2-phenylindole (DAPI). Insets show magnification of the centriole area. Scale bar, 10 μm. **b–d** Histogram showing the percentage of pseudo-bipolar mitosis (centrosome clustering) and multipolar mitosis (non-efficient centrosome clustering) in the indicated cancer cells (**b**) or MDA-MB-231 cells (**c, d**) in response to IR, etoposide (5 μM), or the indicated agents for 15 h. **b** Two-tailed t test p values: p = (MB-231) 0.0086 (Eto), 0.0072 (IR); (BT-549) 0.0074 (Eto), 0.0025 (IR); (HCT116) 0.0068 (Eto), 0.0085 (IR); (H1299) 0.0317 (Eto), 0.0372 (IR); (EJ-1) 0.0066 (Eto), 0.0024 (IR); and (HeLa) 0.0412 (Eto), 0.0305 (IR). **c** T test p values (from left to right): p = 0.0021, 0.0096, 0.0118, 0.0229, 0.0148, 0.0255, 0.0070, 0.0075, and 0.0276. **d** T test p values (from left to right): p = 0.0028, 0.0050, and 0.0012. (**d**) MDA-MB-231 cells were infected with shN or KIFC1-shR virus for 48 h. Representative western blots show the knockdown efficiencies of KIFC1. **e, f** Tissue sections of PDX models (**e**) or MDA-MB-231 xenograft tumors (**f**) were stained with γ-tubulin antibody. DNA was stained with hematoxylin. Representative images showed bipolar (black box), pseudo-bipolar (centrosome clustering, yellow box), and multipolar mitosis (red box) in tissue sections (scale bar, 10 μm). The boxed enlargements showed centrosomes in cells. Histogram showing the percentage of centrosome clustering in tumor sections. **e** T test p values (from left to right): p = 0.0015, 0.0047, 0.0092, 0.0099, 0.0071, 0.0088, 0.0096, and 0.0064. **f** T test p values (from left to right): p = 0.0049, 0.0098, and 0.0143. Statictical data presented in this figure show mean values ± SD of three times of independent experiments. Statistical significance was determined by Two-tailed t test. *p < 0.05, **p < 0.01, Source data are provided as a Source Data file.

upregulation (Fig. 4e). These results indicate that both ATM and ATR are required for DNA damage-induced KIFC1 accumulation.

We next examined whether ATM and ATR kinases phosphorylated KIFC1 after DNA damage. The results showed that endogenous Flag-KIFC1 interacted with ATM and ATR in the presence or absence of etoposide (p-ATM and γH2AX indicate the activation of DNA damage response) (Fig. 4f). Etoposide treatment markedly increased KIFC1 phosphorylation as detected by using a phospho-S/TQ antibody targeting ATM/ATR common substrates (Fig. 4g). LC-MS/MS analysis identified eight phosphorylation sites in KIFC1 (Fig. 4h and Fig. S4), but only S26 conformed to the ATM/ATR substrate S/T-Q motif. We therefore generated a KIFC1 phospho-S26 specific antibody and verified its specificity (Fig. 4i), and further confirmed that etoposide treatment markedly induced endogenous and exogenous KIFC1-S26 phosphorylation (Fig. 4j, k), which could be eliminated by addition of the ATM or ATR inhibitors (Fig. 4k). In addition, the activation of KIFC1-S26 phosphorylation in ATR-sh cells was faster than that in ATM-sh cells (Fig. S5a), indicating that activation of KIFC1-S26 phosphorylation has different dynamics. The activation of ATR by stalled replication forks needed more time compared with the activation of ATM by double-strand breaks after etoposide treatment[28,29]. Other DNA-damaging therapies such as camptothecin (CPT)[36] and ionizing radiation[37] also induced KIFC1-S26 phosphorylation (Fig. S5b, c). Collectively, these results confirmed that both ATM and ATR phosphorylated KIFC1 at S26 after DNA damage.

**KIFC1-S26 phosphorylation promotes centrosome clustering.** To examine the function of KIFC1-S26 phosphorylation in regulating centrosome clustering, endogenous KIFC1 was replaced by wild-type (WT), phosphor-deficient (S26A) or phosphor-mimetic (S26D) Flag-tagged KIFC1 at a dose similar to that of endogenous KIFC1 in MDA-MB-231cells (Fig. 5a) to avoid microtubule bundles and longer spindles, which are induced by KIFC1 overexpression (Fig. S6)[38,39]. Using these cell lines, we observed that the stability of the KIFC1-WT was higher than that of the S26A mutant but was much lower than that of the S26D mutant (Fig. 5b). The S26D mutant showed a decrease, whereas the S26A mutant showed an increase in ubiquitination compared to KIFC1-WT, and etoposide treatment led to a decreased ubiquitination of wild-type but not mutant KIFC1 (Fig. 5c). These results indicate that KIFC1-S26 phosphorylation stabilizes KIFC1 by inhibiting its ubiquitination.

Immunofluorescence analysis showed that the percentage of centrosome amplification in KIFC1-S26A mutant cells was less than that in KIFC1-WT and KIFC1-S26D mutant cells, and was

induced by etoposide treatment at 48 h in all these stable cell lines (Fig. 5d). Knockdown of KIFC1 significantly decreased the frequency of centrosome clustering coupled with increasing of multipolar mitosis, the KIFC1-WT cells showed a significant increase in the frequency of centrosome clustering coupled with a marked decrease of multipolar mitosis, and etoposide increased the frequency of centrosome clustering in cells with wild-type KIFC1 (ShN and KIFC1-WT) but not in cells with KIFC1 mutants (S26A and S26D) (Fig. 5e). ATM/ATR inhibitor VE-822 significantly inhibited etoposide-induced enhancement of centrosome clustering in KIFC1-WT but not in the KIFC1 mutants (S26A and S26D) cells (Fig. 5f). These results suggest that DNA damage treatments promote centrosome clustering in a KIFC1-S26 phosphorylation-dependent manner.

**KIFC1-S26 phosphorylation induces drug resistance.** KIFC1 is also associated with nuclear importins[40,41] and an acentrosomal spindle organization[42]. Thus, we further analyzed the influence of KIFC1-S26 phosphorylation on cell cycle progression (Fig. S7a)[40], nuclear membrane (Fig. S7b)[40], and acentrosomal poles (Fig. S8). KIFC1-S26A cells showed slightly prolonged S and G2/M phases compared with KIFC1-WT and S26D cells (Fig. S7a). There were no significant differences in the degree of DNA damage and in the percentage of the aberrant nuclear membrane in KIFC1-rescued stable cell lines with normal 2 centrosomes or >2 centrosomes (Fig. S7b, c). The percentage of acentrosomal poles in cells with two centrosomes was significantly increased to 30–38% after etoposide treatment (Fig. S8a, d). However, KIFC1-S26A cells with 2 centrosomes showed a significant increase in the frequency of multipolar spindles containing additional pole structures devoid of bona fide centrosomes after DNA damage (Fig. S8a, d). Centrosome clustering is essential for the survival of cancer cells with extra centrosomes. Thus, we concluded that non-efficient centrosome clustering (in cells with extra centrosomes) and acentrosomal poles-induced multipolar spindles (in cells with two centrosomes) in KIFC1-S26A cells might increase sensitivity to etoposide treatment.

We therefore examined the role of KIFC1-S26 phosphorylation in drug resistance by using MDA-MB-231 cells, in which the percentage of centrosome amplification was ~36% and further increased to ~50% after etoposide treatment as shown in Fig. 5d. Increased apoptosis (the activated caspase-3 and the cleaved PARP) was detected in KIFC1-S26A cells (Fig. S8c). KIFC1 knockdown markedly enhanced etoposide-induced cell death, and the KIFC1-S26A mutant-rescued cells showed greater sensitivity to etoposide treatment than KIFC1-WT-rescued and KIFC1-S26D-rescued cells (Fig. 6a). The ATM and ATR inhibitor VE-822 dramatically increased etoposide-induced cell death in

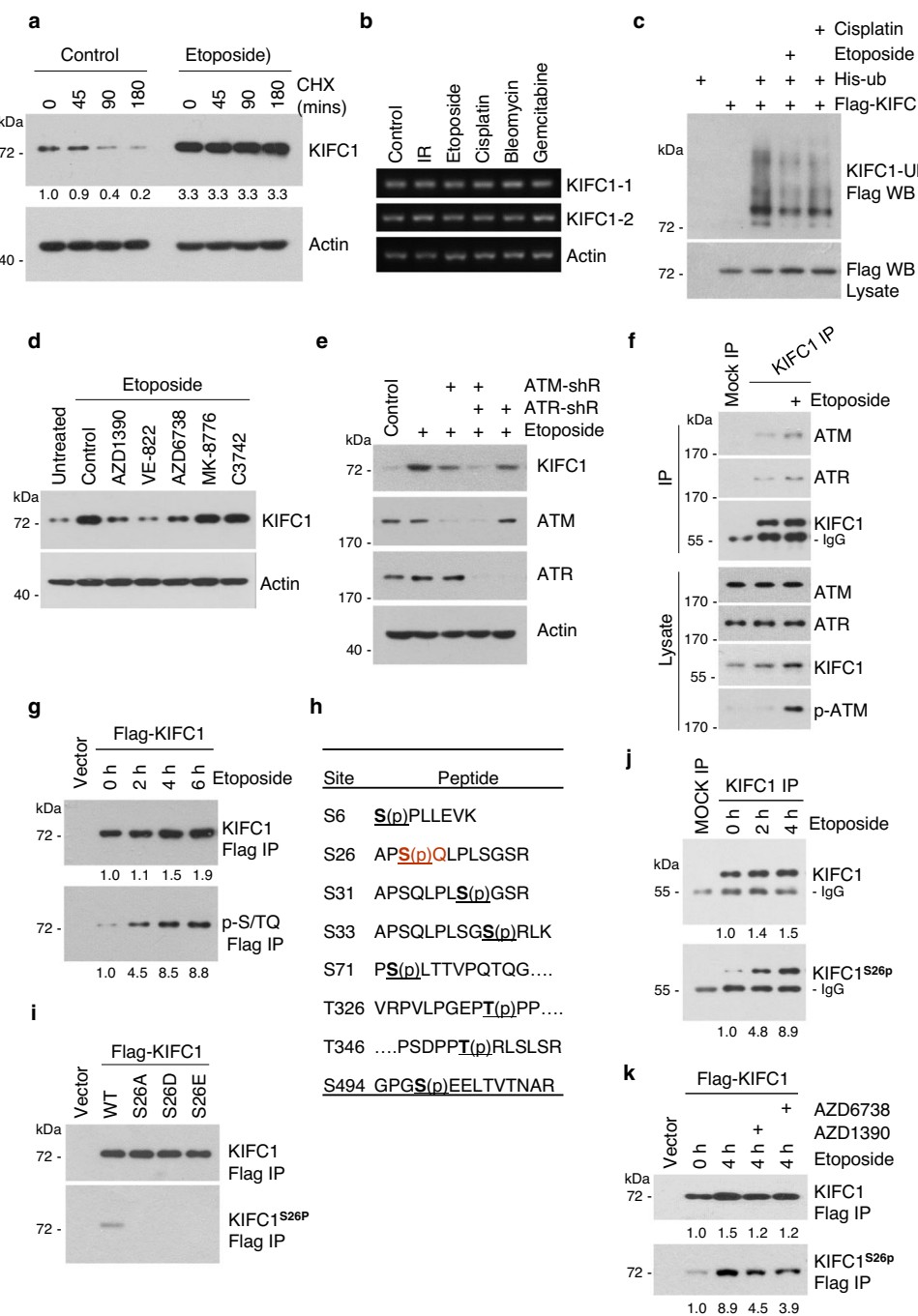

KIFC1-WT, but slightly increased etoposide-induced cell death in the KIFC1 mutants (S26A and S26D) cells (Fig. 6b). Given VE-822 also prevented DNA repair to sensitizes DNA-damaging therapy[34], we conclude that VE-822 sensitizes etoposide treatment via the inhibition of centrosome clustering and prevention of DNA repair. Collectively, ATM/ATR-induced KIFC1-S26 phosphorylation promotes drug resistance.

Next, we investigated the in vivo activity of KIFC1-S26 phosphorylation on drug resistance using the murine tumor xenograft model generated with the KIFC1-WT, KIFC1-S26A, or KIFC1-S26D rescued MDA-MB-231 cells. The results showed that the xenograft tumors with KIFC1-S26A were more sensitive to etoposide than those of KIFC1-WT and KIFC1-S26D (observed by a reduction of tumor volume and weight) (Fig. 6c–e).

The γ-tubulin staining of tumor sections identified that the KIFC1-S26D xenograft tumors showed a significant increase in the frequency of centrosome clustering coupled with a marked decrease of the frequency of multipolar mitosis, and etoposide treatment specifically increased centrosome clustering in the KIFC1-WT but not the KIFC1-S26A/D xenograft tumors (Fig. 6f). Moreover, the ATM/ATR inhibitor VE-822 sensitized xenograft tumors to etoposide treatment (Fig. 6g–i), and analysis of tumor sections showed that VE-822 decreased the frequency of centrosome clustering (Fig. 6j). To further study the inhibitory function of VE-822 on tumor growth, we bore xenograft tumors in nude mice with a HeLa monoclonal cell line with inducible CA (TETON-PLK4 cell line, the percentage of CA was from 11% up to ~80% after doxycycline treatment)[2,43,44], or MDA-MB-231

**Fig. 4 ATM and ATR kinases phosphorylate KIFC1-S26 during DNA-damage conditions. a, b** MDA-MB-231 cells were treated with IR or indicated agents for 15 h. The protein and mRNA levels of endogenous KIFC1 and β-actin (as the internal standard) were examined by western blotting (**a**) and reverse transcription–PCR (**b**). The data are representative of two independent experiments. **a** Cycloheximide (CHX, 50 μm mL$^{-1}$) was added in cells for the indicated time prior to lysis. Relative KIFC1 band intensities were quantified using densitometry and presented. (**c**) 293 T cells transfected with His-ubiquitin and Flag-KIFC1 plasmids were treated with etoposide (20 μM) or cisplatin (20 μM) for 6 h. MG132 (25 μM) was added for 3 h prior to lysis. Ubiquitinated proteins were precipitated using Ni-NTA beads. KIFC1 ubiquitination was detected by western blot using anti-Flag antibody. **d, e** MDA-MB-231 cell lysates were immunoblotted with antibodies against KIFC1, ATM, ATR, or β-actin (as the internal standard). **d** The cells were pretreated with AZD1390 (20 nM), VE-822 (5 μM), AZD6738 (25 nM), MK-8776 (5 μM), or C3742 (10 μM) for 1 h, and then treated with etoposide and these inhibitors for another 15 h. **e** Cells were infected with shN (Control), ATM-shR, or ATR-shR lentivirus for 72 h and then treated with etoposide for another 15 h. **f, j** MDA-MB-231 cells were treated with etoposide (20 μM) for 2 h (**j**) or 4 h (**f, j**). Endogenous KIFC1 was precipitated using anti-KIFC1 antibody or with IgG (Mock IP), and the precipitated proteins were analyzed by western blotting using the antibody against KIFC1, ATM, ATR, p-ATM (S1981), or KIFC1$^{S26p}$. **g** 293T cells transfected with Flag-KIFC1 plasmids were treated with or without etoposide (20 μM) for the indicated time, followed by immunoprecipitation with FLAG-M2 beads. The samples were immunoblotted with antibodies against Flag or p-S/TQ (ATM/ATR$^{sub}$). **h** Eight phosphorylation sites of KIFC1 were identified by LC-MS/MS analysis using purified KIFC1 from Flag-IP in 293 T cells. **i, k** Characterization of KIFC1-S26 phosphorylation antibody (KIFC1$^{S26p}$, produced in our lab). 293T cells were transfected with indicated Flag-KIFC1 plasmids and followed by immunoprecipitation with FLAG-M2 beads. The precipitated proteins were analyzed by western blot KIFC1$^{S26p}$ or Flag antibodies. **k** 293T cells were pretreated with AZD1390 or AZD6738 for 1 h, and then treated with etoposide and these inhibitors for another 4 h. The western blot images are representative of 2 independent experiments with similar results. Source data are provided as a Source Data file.

cells (with high-frequency of CA). PLK4 overexpression and centrosome amplification were markedly induced in the cell line with inducible CA after doxycycline treatment (Fig. S9a, b). VE-822 dramatically inhibited the volumes and weights of tumors of the cells with inducible CA (DOX+) and MDA-MB-231 cells, but not the tumors of the cells with inducible CA (DOX−) and HeLa cells (DOX+) (Fig. S9c–e). These results suggest that VE-822 treatment inhibits the growth of tumors with high-frequency of CA. In tumors with etoposide-induced centrosome amplification, VE-822 sensitizes xenograft tumors to etoposide treatment.

Collectively, these above results demonstrate that KIFC1-S26 phosphorylation promotes the survival of cancer cells with extra centrosomes after treatment with etoposide, and this effect is inhibited by treatment with VE-822 both in vitro and in vivo.

**KIFC1-S26 phosphorylation induces chromosomal instability.** Multipolar mitosis leads to higher levels of aneuploidy than pseudo-bipolar mitosis (centrosome clustering) in tumor cells with supernumerary centrosomes. However, the cells with multipolar mitosis may die due to high levels of aneuploidy, and the progeny of these cells are typically not viable[9]. In contrast, tumor cells undergoing centrosome clustering can survive and further proliferate. With the proliferation of cancer cells, lagging chromosomes induced by centrosome clustering promote CIN[2,9,43,45]. Thus, we next assessed the effect of KIFC1-S26 phosphorylation on the percentage of lagging chromosomes induced by centrosome clustering. Immunofluorescence staining analysis showed that the percentage of lagging chromosomes induced by centrosome clustering in the KIFC1-S26D cell line (~25%) was significantly higher than that in the KIFC1-S26A cell line (~8%). Etoposide treatment led to an increased frequency of lagging chromosomes induced by centrosome clustering in KIFC1-WT and normal control cells, but not in KIFC1 mutants (S26A and S26D) cells. VE-822 treatment markedly decreased etoposide-induced enhancement of the frequency of lagging chromosomes in KIFC1-WT and normal control cells (Fig. 7a). These results indicate that KIFC1-S26 phosphorylation leads to a high tendency for lagging chromosomes induced by centrosome clustering.

To further confirm the relationship between KIFC1-S26 phosphorylation and CIN, we used a non-transformed human mammary epithelial cell line MCF-10A, in which the percentage of centrosome amplification was ~8% and further increased to ~30% after etoposide treatment[2]. The KIFC1 (WT, S26A, and S26D)-rescued MCF-10A cell lines were generated (Fig. 7b), and

we observed similar results to KIFC1-rescued MDA-MB-231 cell lines on the change of centrosome amplification and clustering (Fig. 7c, d). The KIFC1-rescued MCF-10A cell lines were treated with a low-concentration of etoposide for 30 generations, and then the surviving cells were used for assessing the frequency of CIN. The results showed that the percentage of aneuploidy (with more or less than 46 chromosomes per cell) was significantly higher in the KIFC1-WT and KIFC1-S26D cells than that in the KIFC1-S26A cells and untreated MCF-10A cells (Fig. 7e). Fluorescence in situ hybridization analysis of chromosomes 3 and 7 revealed that the KIFC1-WT and KIFC1-S26D cell lines, but not the KIFC-S26A cell line, possessed increased aneuploidy after 30 generations (Fig. 7f). Without etoposide treatment, the percentage of aneuploidy in KIFC1-WT and KIFC1-S26A cells was low and similar to that in untreated MCF-10A cells, indicating that etoposide-induced centrosome amplification and clustering were necessary for the occurrence of CIN (Fig. 7e, f). These results indicate that KIFC1-S26 phosphorylation promotes CIN after treatment with DNA-damaging agents.

**The ATM/ATR-KIFC1-centrosome clustering pathway promotes tumor recurrence.** Centrosome clustering contributes to CIN and leads to tumor heterogeneity thus accelerating the development of malignant characteristics[7,46]. We showed that KIFC1-S26 phosphorylation promoted centrosome clustering and increased the survival of cells with CIN. Therefore we suggest that the surviving cells that undergo centrosome clustering may become potential seed cells for tumor recurrence. To assess this hypothesis, we generated xenograft tumors using the KIFC1-WT, KIFC1-S26A, and KIFC1-S26D rescued MDA-MB-231 cell lines, and determined the tumor growth and recurrence in the presence of etoposide alone and in combination with PBS, VE-822, or CW069 (Fig. 8a). The results showed that the KIFC1-WT xenograft tumors treated with VE-822 or CW069 and the KIFC1-S26A xenograft tumors reached 300 mm$^3$ in ~2 months, whereas KIFC-WT and KIFC-S26D tumors reached the same volume in only ~1 month (Fig. 8b). The rate of CIN after etoposide treatment was significantly higher in the KIFC1-WT and KIFC1-S26D tumors (~40%) than that in KIFC1-S26A cells (~20%) (Fig. 8c). FISH analysis of chromosomes 3 and 7 further confirmed that the KIFC-WT- and KIFC-S26D cells, but not the KIFC-S26A cells, possessed increased CIN (Fig. 8d). After the observation of 150 days post-surgery, a Kaplan-Meier analysis was performed using the presence of a palpable tumor as criteria for tumor recurrence. We found that mice bearing KIFC1-WT tumors (3

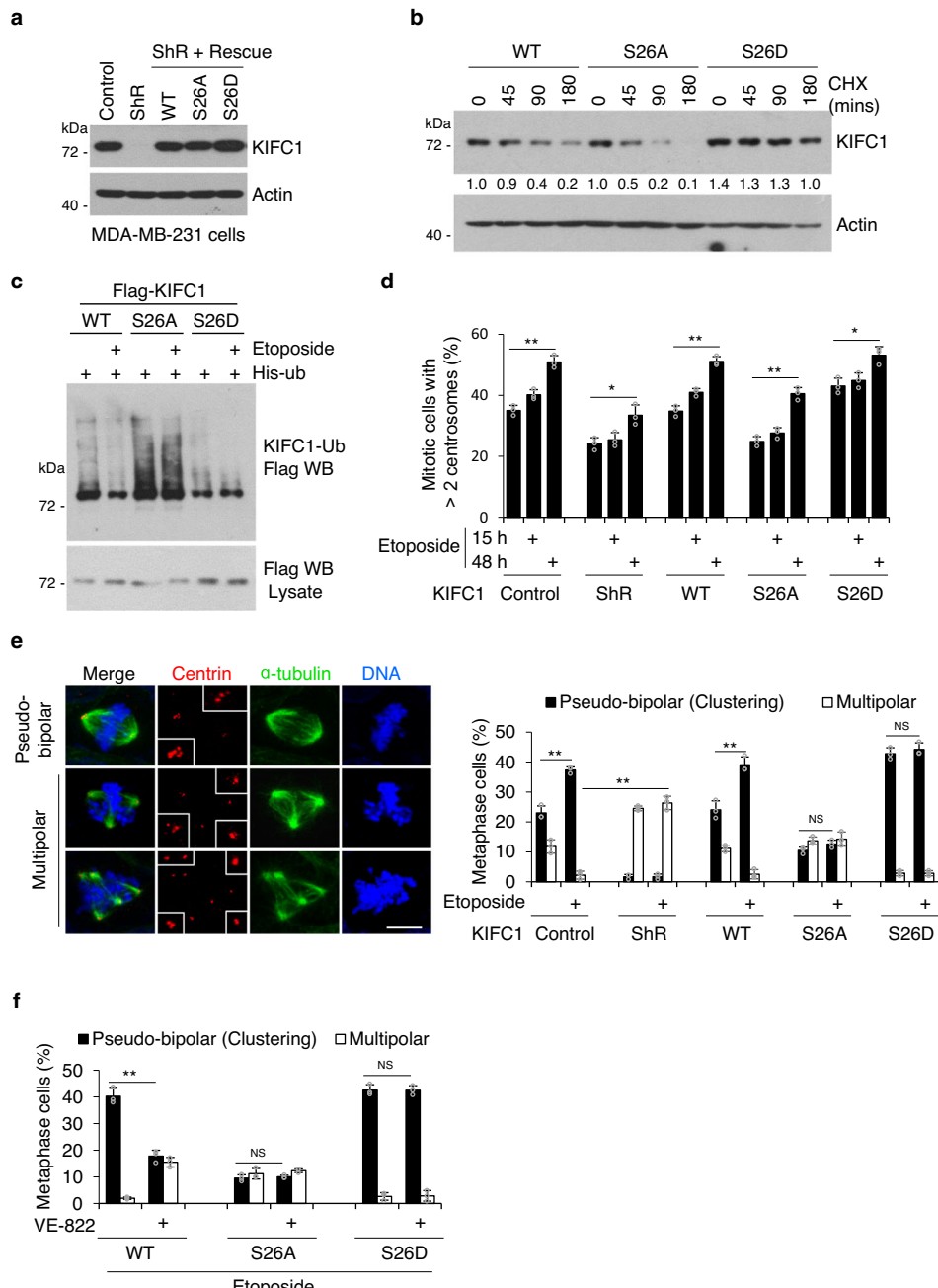

**Fig. 5 KIFC1-S26 phosphorylation promotes centrosome clustering. a** Establishment of Flag-tagged KIFC1-WT, S26A, or S26D mutant stable cell lines from MDA-MB-231 cells. Cell lysates were immunoblotted with antibodies against KIFC1 and β-actin (as internal standard). **b** KIFC1-S26 phosphorylation stabilizes KIFC1 protein level. Western blot analysis of lysates of indicated cell lines treated with cycloheximide (CHX, 50 μm ml$^{-1}$). Relative KIFC1 band intensities were quantified using densitometry and presented. **c** 293T cells transfected with His-ubiquitin and Flag-KIFC1 or indicated Flag-KIFC1 mutant plasmids were treated with etoposide (20 μM) for 6 h. MG132 (25 μM) was added for 3 h prior to lysis. Ubiquitinated proteins were precipitated using Ni-NTA beads. KIFC1 ubiquitination was detected by western blot using anti-Flag antibody. The Western Blot images are representative of 2 independent experiments with similar results (**a–c**). **d–f** KIFC1-S26 phosphorylation promotes centrosome clustering. Histogram showing the percentage of >2 centrosomes per cell (centrosome amplification) (**d**), pseudo-bipolar mitosis (centrosome clustering), and multipolar mitosis (non-efficient centrosome clustering) (**e, f**) in the indicated stable cell lines in response to etoposide (5 μM) for 15 h (**d–f**) or 48 h (**d**). **e** Representative images showing pseudo-bipolar and multipolar mitosis in indicated cell lines (scale bar, 10 μm). Spindle poles, centrioles, and DNA were co-stained with α-tubulin, centrin, and DAPI. Insets show magnification of the centriole area. **f** The cells were pretreated with VE-822 (5 μM) for 1 h and then treated with etoposide and VE-822 for another 15 h. **d** Two-tailed *t* test *p* values (from left to right): *p* = 0.0006, 0.0364, 0.0071, 0.0050, and 0.0441. **e** Two-tailed *t* test *p* values (from left to right): *p* = 0.0073, 0.0019, 0.0034, 0.1816, and 0.6059. **f** Two-tailed *t* test *p* values (from left to right): *p* = 0.0088, 0.3163, and 0.4928. Statistical data show mean values ± SD of three independent experiments. NS = not significant, *$p < 0.05$, **$p < 0.01$. Source data are provided as a Source Data file.

out of 10) and KIFC1-S26D tumors (4 out of 10) showed recurrence in situ for at least 120 days after surgical removal of the original tumor. There was no recurrent tumor in the mice bearing KIFC1-S26A tumors and KIFC1-WT tumors combined treatment with VE-822 or CW069 (Fig. 8e). Moreover, the mice bearing rescue-WT ($n = 2$) and rescue-S26D ($n = 3$) tumors showed distant recurrence to the lung (Fig. 8f). The analysis of chromosome number and FISH also confirmed that tumor cells of recurrence-WT and S26D had higher levels of CIN than untreated MDA-MB-231 cells (Fig. 8g, h). Taken together, the results indicate that KIFC1-S26 phosphorylation drives local and distant tumor recurrence through increased CIN, and either VE-822 or CW069 treatment prevented tumor recurrence after etoposide treatment.

## Discussion

Although radiotherapy and chemotherapy are currently effective methods to treat cancer, therapy resistance and tumor recurrence are still major negative complications. Previous studies revealed that CIN develops drug-resistance and increases the risk of tumor recurrence in cancer therapy[6–8,47,48], and centrosome clustering contributes to CIN[9]. However, the regulatory mechanisms of centrosome clustering and its function in cancer therapy are still poorly understood. In this study, we demonstrated that KIFC1 was a biomarker of recurrence in human breast and colon cancers. DNA-damaging treatments, including radiotherapy and chemotherapies, promoted KIFC1-phosphorylation-dependent centrosome clustering and the survival of cells with extra centrosomes. KIFC1-S26 phosphorylation led to CIN and tumor recurrence both in vivo and in vitro. In addition, we showed that pharmacological inhibition of KIFC1 phosphorylation markedly repressed centrosome clustering and tumor recurrence after chemotherapy. These findings demonstrate that DNA damage-induced KIFC1 phosphorylation promotes centrosome clustering, the survival of cells with CIN, and thus increase the risk of tumor metastasis and recurrence. Notably, the biomarkers that predict

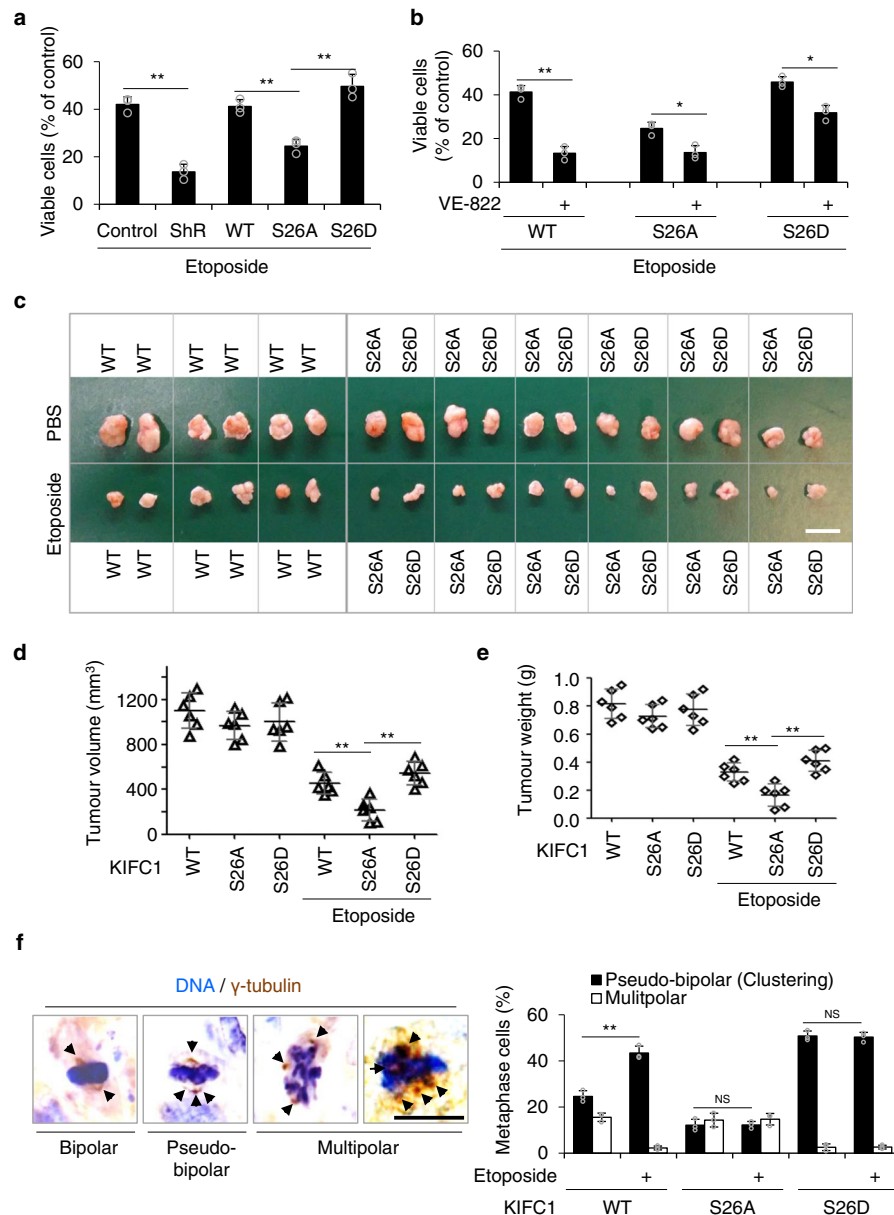

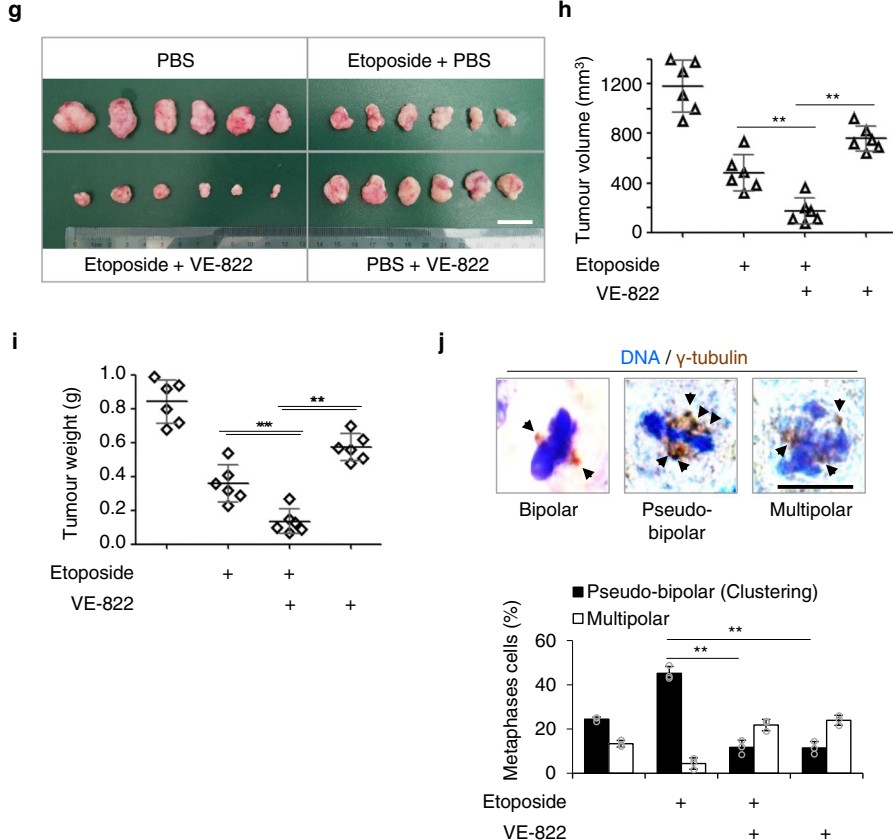

**Fig. 6 KIFC1-S26 phosphorylation induces drug resistance. a, c–f** KIFC1-S26 phosphorylation induces etoposide resistance. **b, g–j** Etoposide-resistance is restrained by ATM/ATR inhibitor VE-822. **a, b** The indicated stable cell lines were treated with etoposide (0.5 μM) or VE-822 (5 μM) for 4 days, and then were analyzed for cell viability using the MTT assay. Histogram graphs showing the percentage of surviving cells. **a** Two-tailed *t* test *p* values (from left to right): *p* = 0.0007, 0.0099, and 0.0090. **b** Two-tailed *t* test *p* values (from left to right): *p* = 0.0005, 0.0110, and 0.0143. **c, g** Xenograft experiment with KIFC1-WT, S26A, or S26D stable cells was described in the Methods section. Tumors were collected and photographed (scale bar, 2 cm). **d, e, h, i** Quantification of average tumor volume (**d, h**) and weight (**e, i**). Six tumors were included in each group. **d** Two-tailed *t* test *p* values (from left to right): *p* = 0.0062 and 0.00001. (**e**) Two-tailed *t* test *p* values: *p* = 0.0099 and 0.00001. **h** Two-tailed *t* test *p* values: *p* = 0.0053 and 0.0016. **i** Two-tailed *t* test *p* values: *p* = 0.0057 and 0.0016. **f, j** Representative immunohistochemical images showing γ-tubulin staining (scale bar, 10 μm) with quantitative analysis of pseudo-bipolar (centrosome clustering) and multipolar mitosis of the tumor sections of xenograft tumor samples after treatment with etoposide or VE-822. Arrows point to the centrosomes. For each experimental condition, 100–161 cells were counted, and three independent experiments were performed. **f** Two-tailed *t* test *p* values: *p* = 0.0016, 0.9245, and 0.5663. **j** Two-tailed *t* test *p* values: *p* = 0.0006 and 0.0092. Data represent the mean ± SD of three times of independent experiments. NS = not significant, *\*p* < 0.05, *\*\*p* < 0.01, Source data are provided as a Source Data file.

clinical response, tumor recurrence, or patient survival are severely lacking for most human cancers. In this study, we not only identified that KIFC1 served as a potential biomarker of recurrence in human breast and colon cancers, but we also showed that DNA-damaging treatments significantly increased the phosphorylation of KIFC1-S26 which leads to the survival of centrosome-amplified cancer cells. It will therefore be intriguing to further explore using KIFC1-S26 phosphorylation as a more specific biomarker for DNA damaging treatment resistance and tumor recurrence.

DNA damage agents such as IR and chemotherapeutic drugs induce DNA lesions, which produce mutations or large-scale genome aberrations. Cancer cells containing DNA lesions activated sophisticated signaling networks that decide cell fate, not only promoting DNA repair and survival but also triggering apoptosis, necrosis, or senescence[49]. ATM and ATR, as central components of networks, phosphorylate a multitude of proteins that either help cells to survive or destine them to undergo cell death[27]. Our study revealed that ATM and ATR induced KIFC1-S26 phosphorylation-dependent centrosome clustering and survival of centrosome-amplified cancer cells after DNA damage.

The surviving cells were characterized with high CIN, invasion, and drug-resistance, and thus served as seed cells for tumor recurrence. The ATM/ATR-derived centrosome clustering of centrosome amplified cancer cells, therefore provides a new way to study tumor cell resistance to DNA damaging therapy, CIN, and tumor recurrence.

Posttranslational modification of proteins is a key step in the process of regulating cellular functions and activities. A previous study reported that CDK1 phosphorylated KIFC1 at Ser6 during mitosis to stabilize KIFC1 protein levels[50]. In this study, we showed that ATM/ATR phosphorylated KIFC1-S26 during DNA-damaging conditions, which led to increased stability of KIFC1 via reducing its ubiquitination and thus promoting its activity in centrosome clustering, suggesting a mechanism for the KIFC1 regulation during stress conditions. In addition, we also identified seven other specific KIFC1 phosphorylation sites, including S6, S31, S33, S71, T326, T346, and S494 in vitro using LC/MS/MS. It will be important to determine whether these modifications affect its cellular and biological functions and activities under normal and stressful conditions. Although CW069 and VE-822 are not specific inhibitors of KIFC1 or

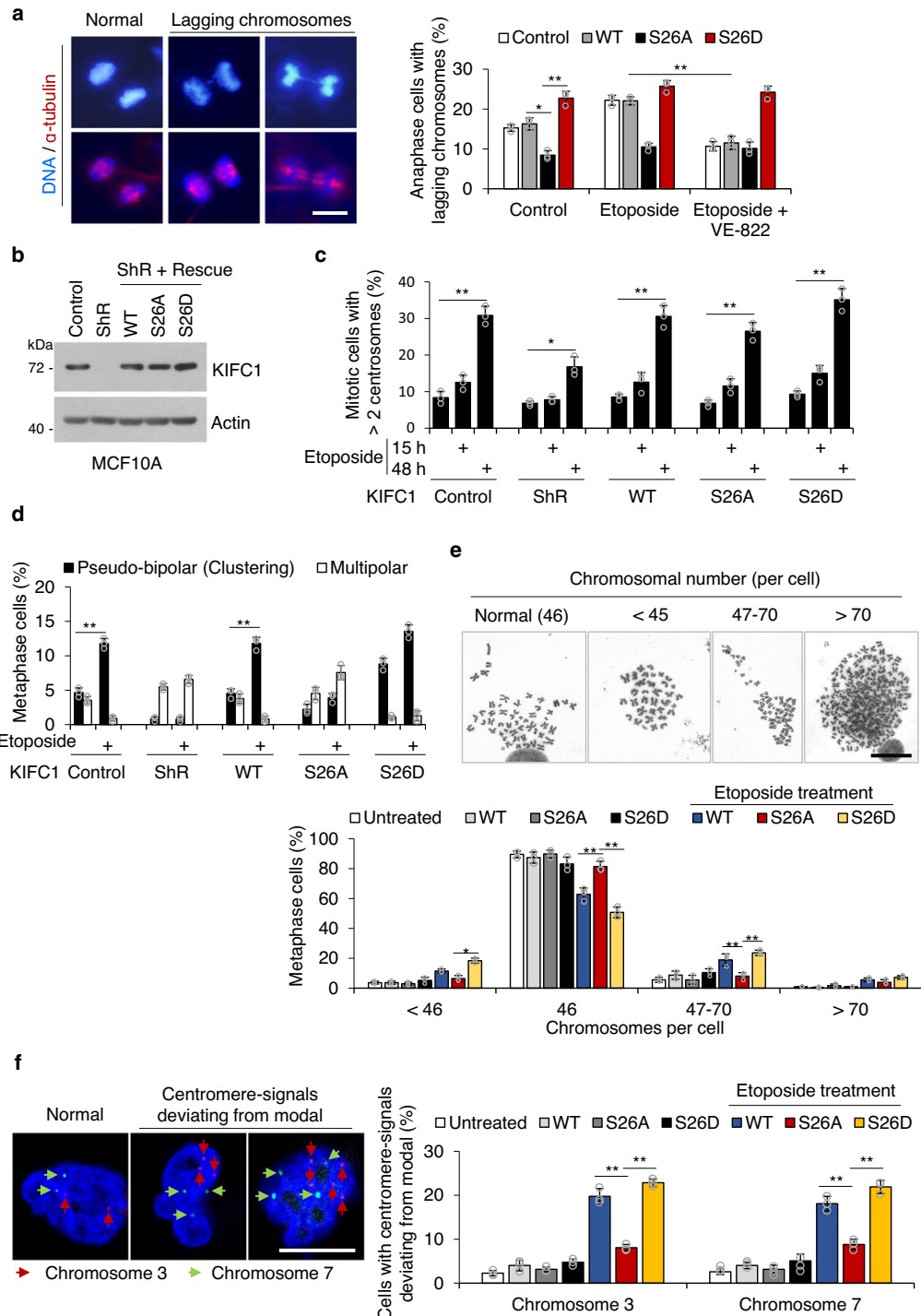

KIFC1-S26 phosphorylation, they both prevent tumor recurrence through KIFC1 inhibition. In this study, they were used to show the significance of KIFC1 or S26 phosphorylation inhibition in DNA-damaging therapies. Developing small molecule compounds to specifically target KIFC1-S26 phosphorylation is a promising therapeutic strategy to address tumor recurrence.

In summary, we suggest that DNA damage-inducing therapies such as IR and chemotherapy drugs not only destroy cancer cells by damaging their DNA, but also trigger an ATM/ATR-KIFC1 phosphorylation-centrosome clustering pathway to selectively maintain the survival of the centrosome-amplified cancer

cells, which in turn leads to CIN, metastasis, and tumor recurrence. Consistent with this model, pharmacological inhibition of KIFC1-S26 phosphorylation specifically induces multipolar cell division and cell death of centrosome-amplified cancer cells via blocking centrosome clustering, thus inhibiting tumor malignant transformation and the rate of recurrence after DNA-damaging therapy. To the best of our knowledge, this is the first study to show that the DNA damage responsive kinases also induce centrosome clustering, which in turn could lead to a high risk of drug resistance and tumor recurrence. Our results provide insights into how tumors acquire the high risk of recurrence and therapeutic

**Fig. 7 KIFC1-S26 phosphorylation induces chromosomal instability. a** Representative images showing lagging chromosomes induced by centrosome clustering in the MDA-MB-231 stable cell lines. The cells were pretreated with VE-822 for 1 h, and then treated with etoposide (2 μM) for another 15 h. Spindle and DNA were co-stained with DAPI and α-tubulin. In quantitative analysis, anaphase cells with lagging chromosomes induced by centrosome clustering were compared with all anaphase cells. For each experimental condition, 100–110 cells were counted, and three independent experiments were performed. Scale bar, 10 μm. Two-tailed $t$ test $p$ values (from left to right): $p = 0.0115$, 0.0030, and 0.0038. **b** Establishment of KIFC1 WT, S26A, or S26D mutant stable cell lines from MCF-10A cells. Cell lysates were immunoblotted with antibodies against KIFC1 or β-actin (as the internal standard). The Western blot images are representative of two independent experiments with similar results. (**c, d**) Histogram showing the percentage of >2 centrosomes per cell (centrosome amplification) (**c**), pseudo-bipolar mitosis (centrosome clustering), and multipolar mitosis (non-efficient centrosome clustering) (**d**) in the indicated stable cell lines in response to etoposide (5 μM) for 15 h (**c, d**) or 48 h (**c**). **c** Two-tailed $t$ test $p$ values: $p = 0.00044$, 0.0137, 0.0033, 0.0033, and 0.0023. (**d**) Two-tailed $t$ test $p$ values: $p = 0.0097$, and 0.0099. (**e, f**) The stable cell lines were treated with etoposide (0.1 μM) for 30 generations and then were used to assess the rate of chromosomal instability (CIN). For each experimental condition, 100–120 cells were counted, and three independent experiments were performed. Scale bar, 10 μm. **e** Representative metaphase plates containing different chromosome numbers with quantitative analysis of chromosome numbers. For each experimental condition, 100–164 cells were counted, and three independent experiments were performed. Two-tailed $t$ test $p$ values (from left to right): $p = 0.0278$, 0.0078, 0.0098, 0.0079, and 0.0056. **f** Representative image of the centromeric DNA of chromosomes 3 and 7 obtained by fluorescence in situ hybridization (FISH) analysis with quantitative analysis of chromosome numbers. For each experimental condition, 100–112 cells were counted, and three independent experiments were performed. Two-tailed $t$ test $p$ values (from left to right): $p = 0.0027$, 0.0037, 0.0086, and 0.0084. Statistical data presented in this figure show mean values ± SD of three times of independent experiments. *$p < 0.05$; **$p < 0.01$. Source data are provided as a Source Data file.

resistance via activating centrosome clustering, and reveal that blocking KIFC1 phosphorylation may serve as a promising therapeutic strategy for reducing the risk of tumor metastasis and recurrence.

## Methods

**Cell lines, plasmids, and reagents**. MDA-MB-231, HCT 116, MCF-10A, BT549, H1299, 293T, and HeLa were obtained from the American Type Culture Collection (Manassas, VA, USA). OPM-2 cells were obtained from Biovector Science Lab, Inc (Beijing, China). Full-length Wild-type (WT) and site-specific mutants of KIFC1 were cloned into the pCDNA 3.0 vector or pLVX-IRES (lentiviral expression vector) by standard cloning methods[51]. The pLKO.1 control (shN), was generated with control oligonucleotide GCTTCTAACACCGG-AGGTCTT. KIFC1-shR was generated with AACGTTGGACCAAGAGAACCA. ATM-shR was generated with TGGTGCTATTTACGGAGCT. ATR-shR was generated with AAGCGGCCTGAT TCGAGATCCT. WT, S26A, and S26D KIFC1 plasmids contained underlined mutated nucleotides (AACTCTCGATCAGGAAAATCA) to generate shRNA-resistant KIFC1 mutants. VAD1390, VAD6738, VE-822, MK-8776, C3742, etoposide, cisplatin, oxaliplatin, mitomycin C, estramustine, epirubicin, gemcitabine, bleomycin, and CTX were obtained from MedChemExpress (Monmouth Junction, NJ, USA). Doxycycline and cyclohexane were obtained from Sigma-Aldrich (St. Louis, MO, USA). Antibodies against KIFC1 (1:2000, HPA055997, Sigma-Aldrich; 1:500, ab172620, Abcam), phospho-ATM/ATR substrate (S/TQ) (1:500, #9607; Cell Signaling Technology, Danvers, MA, USA), ATM (1:500, ab32420; Abcam, Cambridge, UK), ATR (1:500, YT0416; Immunoway, Plano, TX, USA), p-ATM (1:2000, #5883; Cell Signaling Technology), γH2AX (1:2000, ab26350; Abcam), Flag (1:5000, F1804; Sigma-Aldrich), γ-tubulin (1:500, T5326; Sigma-Aldrich), Centrin (1:500, C7736; Sigma-Aldrich), α-tubulin (1:1000, T5199; Sigma-Aldrich), Lamin A/C (1:300, ab108922; Abcam), Poly (ADP-ribose) polymerase (1:500, 13371-1-AP; Proteintech), activated-caspase-3 (1:500, BS7004; Bioworld Technology), and β-actin (1:5000, A5316; Sigma-Aldrich) were used.

**Tumor-bearing mouse models**. MDA-MB-231 cells or the indicated stable cell lines ($3 \times 10^6$ cells) were subcutaneously injected into the left flank or into both flanks of female BALB/c nude mice (6 weeks old). When the tumors reached 150 mm³, the mice were randomly divided into the control, etoposide, VE-822, or CW069 treatment groups. Etoposide (20 mg/kg per week) or CW069 (200 mg/kg, 4 consecutive days per week) was administered by intraperitoneal injection. VE-822 (20 mg/kg, 4 consecutive days per week) was administered by oral gavage. The tumor size was measured every 5 days using calipers. In Fig. 7, the tumors were extracted from nude mice and analyzed at day 40. In Fig. S9, the indicated groups were fed chow ad libitum containing Doxycycline at 600 mg/kg (Harlan Teklad Diets), and the tumors were extracted from nude mice and analyzed at day 45. In Fig. 8, when the tumor volumes reached 300 mm³, the tumors were surgically removed, and the mice were treated with the indicated drugs for another 2 weeks. After 5 months with no drug treatment, mice with tumor recurrence were identified and analyzed. Animals were treated according to high ethical and scientific standards with oversight from the animal center at East China Normal University.

**Tissue microarray and IHC staining**. Human tissue microarrays of breast cancer (HBreD140Su04; Shanghai Outdo Biotechnology, Shanghai, China) and colorectal cancer (COC1601; Shanghai Superbiotek Pharmaceutical Technology, Shanghai, China) were purchased. The clinical characteristics of all samples were downloaded from the company web sites. Antibodies against KIFC1 were used for

immunohistochemistry staining. The intensity of KIFC1 staining was quantified, scored, and graded (low, 0–4 points; medium, 5–8 points; and high, 9–12 points) as described previously[6]. To ensure an unbiased result, the data were collected in a double-blinded manner. Source data are provided as a Source Data file.

**Patient-derived xenograft (PDX) models**. Breast cancer PDX models were generated by Shanghai LIDE Biotechnology (Shanghai, China) with informed consent from the patients. The experiment procedure was approved by the China Ethics Committee of Registering Clinical Trials (Registration number: ChiCTR1900027396). The clinical characteristics of tumors used in this study are shown in Fig. S2. Each tumor tissue was cut to 3–5 mm tumor masses, and implanted at 4 subcutaneous points in each BALB/c nude mice (6–8 weeks of age) ($n = 6$). Tumor growth was observed daily. When the tumor volume reached 200 mm³, PBS (200 μl), etoposide (40 mg/kg), or cisplatin (20 mg/kg) was administered by intraperitoneal injection. After 24 h, tumors were extracted from the mice. Sections of paraffin-embedded tumor tissues were stained using IHC and visualized by microscopy.

**Reverse transcription-PCR**. RT-PCR analysis was performed as previously described[6]. The following PCR primers were used: KIFC1-1, (5'GGTGCAACGA CCAAAATTACC and 5'GGGTCCTGTCTTCTTGGAAAC), KIFC1-2, (5'TTACA AGTCGTCGCACCTCAA and 5'TCTGGATGATAGGTTGGGTGG) and β-actin, (5'TCCTGTGGCATCCACGAA and 5'TCGTCATACTCCTGCTTGC).

**Fluorescence in situ hybridization**. FISH analysis was performed following the User Manual of the ZytoLight SPEC CDKN2A/CEN 3/7/17 Quadruple Color Probes (Zytovision, Bremerhaven, Germany). Briefly, cells on coverslips were fixed in Carnoy's fixative solution, washed in 2× SSC (2 min), dehydrated in ethanol, treated with DNA probes, and then denatured at 75 °C (2 min). The coverslips were then incubated overnight at 37 °C, washed with 4× SSC (with 0.05% Tween 20, 5 min), incubated in 0.25× SSC at 72 °C (2 min), washed in 4× SSC (with 0.05% Tween 20, 30 s), and then stained with 4′,6-diamidino-2-phenylindole (DAPI). Chromosomes 3 (red), 7 (green), and DNA (blue) were analyzed using confocal microscopy.

**Establishment of stable cell lines**. Using the lentiviral system, MDA-MB-231 and MCF-10A cells were infected with KIFC1-shRNA (selected with 0.5 mg/ml puromycin) and shRNA-resistant KIFC1 WT, S26A, and S26D (selected with 1 mg/ml hygromycin) viruses. The monoclonal WT and mutant cells expressed proteins at a level similar to that of endogenous KIFC1.

Inducible HeLa Tet-on cells stably expressing Flag-PLK4 were successively infected with pLVX-Tet3G virus and pLVX-TRE3G-PLK4 virus and selected with the indicated resistance markers following the use of the Lenti-X™ Tet-On® 3G Inducible Expression System (Takara Bio Inc.)[52]. For the induction of Flag-PLK4 expression, media were supplemented with 2 μg/ml doxycycline (DOX). To select cell lines with induced CA (DOX+) and no induced CA (DOX−), highly effective monoclonal cell lines were chosen.

**Immunofluorescence**. Immunofluorescence was performed as previously described[6,53]. Briefly, cells were fixed with cold methanol, permeabilized, incubated with primary antibodies, and followed with secondary antibodies. Fluorescent secondary antibodies were purchased from Invitrogen (Carlsbad, CA, USA) (1:800, A11004)and Jackson ImmunoResearch (West Grove, PA, USA) (1:600, 111-295-003; 1:600, 111-545-003; 1:600, 115-545-003).

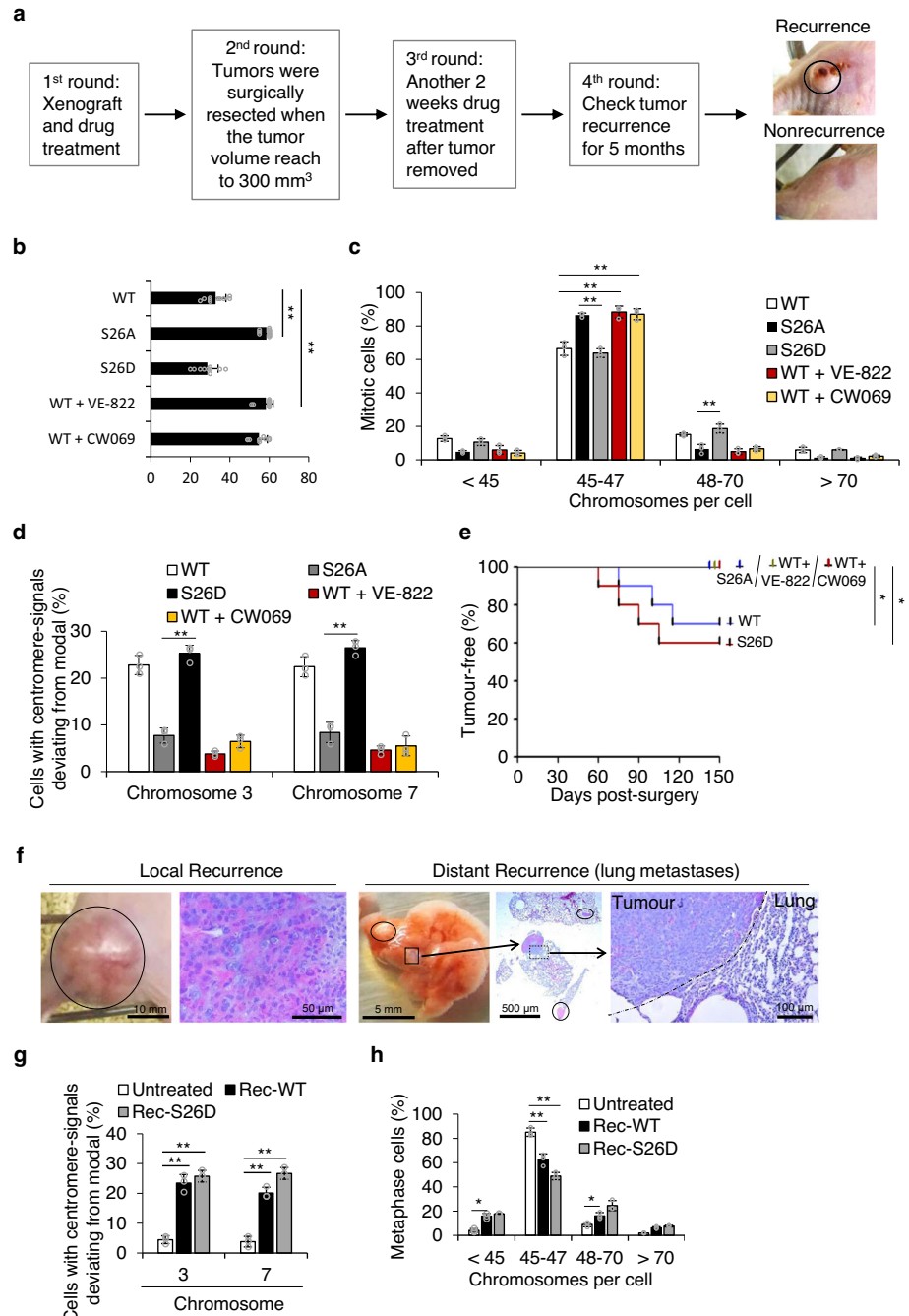

**Immunoprecipitation and western blotting**. Cell lysate preparation and western blotting were performed as previously described[52]. Briefly, cells were lysed in lysis buffer (50 mM Tris-HCl pH 8.0, 5 mM EDTA, 150 mM NaCl, 0.5% NP-40, 1 mM PMSF), centrifuged for 5 min at 10,000 g, and further analyzed with SDS-PAGE and western blotting. For co-immunoprecipitation, cell lysates were immunoprecipitated with anti-Flag-M2 agarose or Protein A/G agarose plus anti-KIFC1 antibody for 4–6 h at 4 °C. The beads were washed extensively with lysis buffer, boiled in SDS sample buffer, fractionated by SDS-PAGE, and analyzed by western blotting. Source data are provided as a Source Data file.

**Aneuploidy analysis**. These experiments were performed as previously described[6]. Briefly, cells were treated with colcemid (50 ng/ml, 37 °C, 6 h), collected, suspended in KCl (75 mM, 37 °C, 15 min), fixed in Carnoy's solution for 30 min, then dropped onto slides and stained with 5% Giemsa solution. The chromosome number was analyzed using confocal microscopy.

**Cell-cycle analysis**. Cells were trypsinized, harvested, and fixed with 0.5 ml 70% EtOH (−20 °C) for 24 h. Cells were incubated with PBS containing 20 µg/ml Propidium Iodide (PI) and 10 µg/ml RNase A for 30 min in 37 °C, and analyzed

using a BD FACSCalibur flow cytometer. Cell cycle fractions were evaluated by automatic analysis using the ModFit LT3.2 software.

**Statistics and reproducibility**. Statistical analysis was carried out using two-tailed Student's *t* test, and the results are expressed as the mean±standard deviation (SD) of three independent experiments. *P* values are indicated in the Figure legends. Microscopy images shown are representative of at least 5 fields from 3 independent experiments. Western Blot images are representative of 2 independent experiments. All biological and biochemical experiments were performed with appropriate internal negative and/or positive controls as indicated.

**Reporting summary**. Further information on research design is available in the Nature Research Reporting Summary linked to this article.

## Data availability
All original data are available upon request. All the other data supporting the findings of this study are available in the article and Supplementary Information files. Source data are provided with this paper. No datasets were generated or analyzed during this study.

**Fig. 8 The ATM/ATR-KIFC1-centrosome clustering pathway promotes tumor recurrence. a** Schematic drawing of tumor recurrence, as described in the Methods section. Xenograft tumors were treated with etoposide alone and in combination with PBS, VE-822, or CW069, and were allowed to grow up to a mean volume of 300 mm³ before being surgically resected. After surgery, the mice were treated with the indicated drugs for another 2 weeks. After 5 months with no drug treatment, mice with tumor recurrence were identified and analyzed. **b** The time at which the indicated tumors reached 300 mm³. Data represent mean ± SD ($n = 10$). Two-tailed $t$ test $p$ values (from left to right): $p = <0.00001$ and $<0.00001$. **c, d** KIFC1-S26 phosphorylation promotes CIN in xenograft tumors. The indicated cells were isolated from the surgically resected tumors and then were cultured to assess the rate of CIN in vitro. For each experimental condition, 100–150 cells were counted, and three independent experiments were performed. **c** The graph shows the fraction of indicated cells with different numbers of chromosomes per cell. For each experimental condition, 100–127 cells were counted, and three independent experiments were performed. Two-tailed $t$ test $p$ values (from left to right): $p = 0.0022$, 0.0035, 0.0021, and 0.0008. **d** The graph shows the fraction of the indicated cells stained for centromeric DNA on chromosomes 3 and 7 with FISH analysis. For each experimental condition, 100–124 cells were counted, and three independent experiments were performed. Two-tailed $t$ test $p$ values (from left to right): $p = 0.0092$ and 0.0077. **e** Graph depicting the Kaplan-Meier analysis of tumor recurrence in the different groups. Survival cutoff criteria (compassionate euthanasia), when the recurrent tumors impeded ambulation, defecation, urination, or eating. Each group, $n = 10$. Two-tailed $t$ test $p$ values (from left to right): $p = 0.0442$ and 0.0291. **f** Representative images showing local recurrence and distant recurrence (lung metastases). The boxed enlargements showed tumor morphology. Hematoxylin and eosin staining of tumor tissue sections. **g, h** KIFC1-S26 phosphorylation promotes CIN in the locally recurrent tumors. Recurrence-WT (Rec-WT) and recurrence-S26D (Rec-S26D) cells were isolated from the indicated recurrent tumors and then were re-cultured to assess the rate of CIN in vitro. The untreated cells were MDA-MB-231 cells. Data represent the mean ± SD of three times of independent experiments. For each experimental condition, 100–109 cells were counted, and three independent experiments were carried out. **g** The graph shows the fraction of indicated cells with different numbers of chromosomes per cell. For each experimental condition, 100–109 cells were counted, and three independent experiments were performed. Two-tailed $t$ test $p$ values (from left to right): $p = 0.0022$, 0.0031, 0.0079, and 0.0024. **h** The graph showed the fraction of the indicated cells stained for centromeric DNA on chromosomes 3 and 7 with FISH analysis. For each experimental condition, 100–159 cells were counted, and three independent experiments were performed. Two-tailed $t$ test $p$ values (from left to right): $p = 0.0261$, 0.0098, 0.0047, and 0.0164. *$p < 0.05$; **$p < 0.01$. Source data are provided as a Source Data file.

The data that support the findings of this study are available from the corresponding author upon reasonable request. Source data are provided with this paper.

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

## Acknowledgements

This work was supported by the National Key Research Program of China (2016YFC1304802), the National Natural Science Foundation of China (31671462, 31970720, 31671433, 31970736, 31970691, 31700691, 32070770), the Shanghai Munici-pal Health Commission (2018YQ44), and the Science and Technology Commission of Shanghai Municipality (19QA1407100 and 16YF1409100).

## Author contributions

G.F., L.S., L.M., C.H., X.W., Z.S., C.H., Y.H., and Q.Y. performed experiments. S.Z. and C.W. interpreted data and wrote the manuscript. L.C., X.Z., Y.Z., X.S., S.X., and B.H. supervised the study and reviewed the manuscript.

## Competing interests

The authors declare no competing interests.
