## [Peer Review File · Nature Communications]

Reviewers' Comments:

Reviewer #1:

Remarks to the Author:

This manuscript identifies KIFC1 phosphorylation as a key event in centrosome clustering and amplification, linking ATM and ATR to this directly. This is novel and worthy of publication because ATM and ATR are druggable targets with clinical agents available to interfere with this tumour-promoting potential.

However, the study requires specific ATM and ATR pharmacological interventions to prove this concept as currently only very poor agents such as non-specific caffeine and early probes have been utilised. Academic labs must stop using caffeine for probing ATM and ATR biology. Exquisitely potent and selective tools such as KU60019, Ku55933 or better still, AZD1390 and AZD6738 should be used. These latter agents can be dosed orally for in vivo xenograft PD-efficacy studies. In the introduction, could the authors also clarify why KIFC1 inhibitors were toxic? Was this on-target tox or off-target tox? If former, then there is little translational use to using ATM or ATR inhibitors as they too, if being used to block KIFC1 phosphorylation as the main goal in blocking tumour cell growth, may also be causing normal tissue toxicities. A comment here would be useful. Mechanistically could the authors provide evidence that ATM can phosphorylate KIFC1 at different or similar dynamics to ATR-mediated phosphorylation? This is because ATR is activated after replication fork stalling and collapse which can take longer than double strand breaks induced by DNA damaging agents such as radiation. The authors use etoposide, a topoisomerase inhibitor, but could the authors probe this further by using other DNA damaging agents to understand the relative need for replication stress?

If these can be addressed, I believe the novelty, the robustness of the data provided and the excellent way in which it is presented, warrants publication in Nature Communications.

Reviewer #2:

Remarks to the Author:

Clustering of extra centrosomes during mitosis is critical for the survival of cancer cells with amplified centrosomes. This makes centrosome clustering an attractive therapeutic target. The kinesin KIFC1/HSET is the major regulator of this process and previous work demonstrated that KIFC1 depletion leads to multipolar mitosis and cell death specifically in cells with amplified centrosomes. However, how this motor is regulated has remained unclear.

In this manuscript, Fan et al described for the first time that activation of the kinases ATM/ATR downstream of DNA damage leads to KIFC1 stabilization and efficient clustering. Mechanistically, ATM/ATR phosphorylates KIFC1 in response to DNA damage leading to its stabilisation. As a consequence, cancer cells treated with DNA damage agents have increased clustering efficiency, which the authors propose could lead to tumour recurrence. They also highlight that blocking KIFC1 phosphorylation, by inhibiting ATM/ATR, in combination with DNA damage agents could be a possible therapeutic avenue.

Overall, I find this manuscript interesting and novel. The experiments are well conducted and controlled and it would be of interest to the broad readership of Nature Comms. However, while the mechanistic studies are clear, the cancer part is less so and overall the conclusions are not well supported by that in the sense that other explanations are likely to exist. Furthermore, the KIFC1 mutants should be well characterised in the context of normal centrosome numbers as well to determine if other effects exist. I would advise the authors to tone down their conclusions or add extra data to fully prove their points. Below I highlight issues that need to be addressed prior to publication:

#1. Figure 4F. exogenous KIFC1 does not seem to get stabilised in response to DNA damage, although it gets phosphorylated. Can the authors explain why that is the case? A lot of the work

they do is based on exogenously expressed kifc1.

#2. Figure 4G. what about interaction with ATM/ATR upon DNA damage? As for the exogenous kifc1 that does not affect but for the endogenous thigs might be different.

#3. Figure 4. Overall it would be helpful to add quantifications to the western blot data. For example 4H.

#4. The identification of the phospho site and phospho-kifc1 antibody data is very nice. However, it will be critical to demonstrate that this is also observed in endogenous kifc1. I don't think the authors show that. Given my point #1 above, it is important to know if this site is also phosphorylated in endogenous protein in response to DNA damage.

#5. Some controls are missing throughout this manuscript. It is essential that the authors show the percentage of centrosome amplification upon treatments, overexpressions etc. Not only the clustering versus multipolar phenotypes. Do the DNA damage treatments used here really induce bona fide centrosome amplification? what about cell lines that express the mutant S26A? in principle these would not cluster well and thus I would expect that the percentage of centrosome amplification would decrease overtime.

#6. One of my main criticisms is the interpretation of the tumour data. there is no proof that the S26A mutants are more sensitive to etoposide due to inefficient clustering. We do not know the percentage of centrosome amplification in any of the conditions.

#7. I am not convinced by the conclusion that the combination of VE-822+etoposide has anything to do with centrosome amplification and clustering. Hasn't it been shown that DNA damage agents in combinations with agents that prevent DNA repair lead to better killing of tumour cells? There is no data to support that this combination is working via preventing centrosome clustering, even if clustering is inhibited. One way to assess that would be to do the combination therapy in the S26D mutant cells. But even in this case a better characterisation of this mutant in cells with normal centrosomes would be necessary.

#8. This brings me to another crucial point. It will be important to describe the role of these kifc1 mutants in cells with normal centrosomes. Kifc1 is dispensable to mitosis in cells with normal centrosomes but when overexpressed is leads to microtubule bundles in interphase. The authors should show images of interphase cells and a characterisation of mitosis in cells with normal centrosomes when these mutants are expressed.

#9. Related with point #8 above, the CIN data in Figure & suggests that these mutants might be doing something else n mitosis leads to increase lagging of chromosomes. This is more evident in 7C where authors use MCF10A cells that have normal centrosome number. What is the actual % centrosome amplification upon etoposide treatment in these cells? And why aren't the authors quantifying these defects in WT, S26A and S26D in the absence of etoposide? This is crucial to know if the effects depend on drug treatment and centrosome amplification. Chromosome segregation should be quantifying in combination with centrin staining to assess whether its only visible in cells with extra centrosomes.

#10. Figure 8E it is impossible to see all the treatments in the graph.

#11. Figure 8. Did the authors assess the toxicity of CW069? They mention that this inhibitor causes unspecific toxicity, as it has been reported. Thus, the use of this inhibitor here is confusing. I am also intrigued to why if clustering is prevented by VE-822 and CW069 more aneuploidy is not observed? Preventing clustering does not immediately kill cells, it takes in fact several days, so not sure I understand this.

#. On page 2: "... either die to chromosome segregation...". This is not correct. As the authors mentioned, centrosome clustering allows survival while leading to chromosome missegregation. What the authors should say instead that high levels of aneuploidy generated by multipolar mitosis is the cause of cell death.

#. On page 2: what do the authors mean by "exhibit a period of division"? this is very unclear to me.

June 28, 2020

Re: Revision to *Nature Communications*

Dear Reviewers:

We deeply appreciate the time and effort you have spent in reviewing our manuscript. The following is a point-by-point response to the questions and comments.

Arrangement of Figures

Original	In Revision
Fig. 4D	Fig. 4D (new, we added new inhibitors and removed caffeine)
Fig. 4F	Fig. 4F (new, the endogenous KIFC1 immunoprecipitation)
Fig. 4G	Removed
Fig. 4H	Fig. 4G (relocated and added quantifications)
Fig. 4I	Fig. 4H (relocated)
Fig. 4J	Fig. 4I (relocated) Fig. 4J (new)
Fig. 4K	Fig. 4K (new, we added new inhibitors and removed caffeine) Fig. 5D (new)
Fig. 5D	Fig. 5E (relocated)
Fig. 5E	Fig. 5F (relocated and removed the samples using caffeine)
Fig. 6B	Fig. 6B (revised) Fig. 7C (new) Fig. 7D (new)
Fig. 7C, D	Fig. 7E, F (revised and relocated)
Fig. 8E	Fig. 8E (showed all the treatments) Supplemental Fig. S3 (new)
Supplemental Fig. S3	Supplemental Fig. S4 (relocated) Supplemental Fig. S5 (new) Supplemental Fig. S6 (new)

Responses to critiques (reviewers' comments are underlined):

Reviewer 1

This manuscript identifies KIFC1 phosphorylation as a key event in centrosome clustering and amplification, linking ATM and ATR to this directly. This is novel and worthy of publication because ATM and ATR are druggable targets with clinical agents available to interfere with this tumour-promoting potential.

However, the study requires specific ATM and ATR pharmacological interventions to prove

this concept as currently only very poor agents such as non-specific caffeine and early probes have been utilised. Academic labs must stop using caffeine for probing ATM and ATR biology. Exquisitely potent and selective tools such as KU60019, Ku55933 or better still, AZD1390 and AZD6738 should be used. These latter agents can be dosed orally for in vivo xenograft PD-efficacy studies.

Response: Thank you for your suggestion. AZD1390 and AZD6738 are appropriate to be used in the study of cancer therapy. We have removed caffeine and repeated the relevant experiments with AZD1390 and AZD6738, as shown in the new Fig. 4D, Fig. 4K, Fig. 5F, Fig. 6B, and Fig. 7A.

In the introduction, could the authors also clarify why KIFC1 inhibitors were toxic? Was this on-target tox or off-target tox? If former, then there is little translational use to using ATM or ATR inhibitors as they too, if being used to block KIFC1 phosphorylation as the main goal in blocking tumour cell growth, may also be causing normal tissue toxicities. A comment here would be useful.

Response: We appreciate your thoughtful comments and suggestions. The KIFC1 inhibitor, CW069, was computationally designed based on the inhibitors of kinesin Eg5 [1], and has off-target toxicity. Another inhibitor, AZ82, effectively bound to the minus end-directed KIFC1. However, a nonspecific cytotoxic effect of AZ82 at concentrations > 4 μ M was found, which prevented further studies regarding whether it could selectively kill cancer cells with amplified centrosomes [2]. Both CW069 and AZ82 showed off-target toxicity, and knockdown of KIFC1 did not affect cellular metabolism and cell division in somatic cells with normal centrosomes [3]. Thus, specifically targeting KIFC1 in DNA-damaging therapy is viable. We have added a more substantial Discussion section in the revised manuscript (Discussion, page 11).

Mechanistically could the authors provide evidence that ATM can phosphorylate KIFC1 at different or similar dynamics to ATR-mediated phosphorylation? This is because ATR is activated after replication fork stalling and collapse which can take longer than double strand breaks induced by DNA damaging agents such as radiation. The authors use etoposide, a topoisomerase inhibitor, but could the authors probe this further by using other DNA damaging agents to understand the relative need for replication stress?

Response: Thank you for your thoughtful suggestion.

First, we examined KIFC1-S26 phosphorylation in ShN (Control), ATM-sh, and ATR-sh cells in the new Fig. S5A. The results showed that the increased KIFC1-S26 phosphorylation in ATR-sh cells occurred more quickly than in ATM-sh cells after etoposide treatment. The activation of ATR by stalled replication forks needed more time compared with the activation of ATM by double-strand breaks after etoposide treatment [4, 5]. Thus, we concluded that ATM phosphorylated KIFC1 at different dynamics to ATR-mediated phosphorylation.

Second, other DNA-damaging therapies such as camptothecin treatment (activated ATR and was related to replication stress) and ionizing radiation also induced KIFC1-S26 phosphorylation (Fig. S4B, C). These results indicate that DNA damage, including double strand breaks and replication fork stalling and collapse, are both significant to KIFC1-S26 phosphorylation.

Reviewer 2

Clustering of extra centrosomes during mitosis is critical for the survival of cancer cells with amplified centrosomes. This makes centrosome clustering an attractive therapeutic target. The kinesin KIFC1/HSET is the major regulator of this process and previous work demonstrated that KIFc1 depletion leads to multipolar mitosis and cell death specifically in cells with amplified centrosomes. However, how this motor is regulated has remained unclear.

In this manuscript, Fan et al described for the first time that activation of the kinases ATM/ATR downstream of DNA damage leads to KIFC1 stabilization and efficient clustering. Mechanistically, ATM/ATR phosphorylates KIFC1 in response to DNA damage leading to its stabilisation. As a consequence, cancer cells treated with DNA damage agents have increase clustering efficiency, which the authors propose could lead to tumour recurrence. They also highlight that blocking KIFC1 phosphorylation, by inhibiting ATM/ATR, in combination with DNA damage agents could be a possible therapeutic avenue.

Overall, I find this manuscript interesting and novel. The experiments are well conducted and controlled and it would be of interest to the broad readership of Nature Comms. However, while the mechanistic studies are clear, the cancer part of less so and overall the conclusions are not well supported by that in the sense that other explanations are likely to exist. Furthermore, the kifc1 mutants should be well characterised in the context of normal centrosome numbers as well to determine if other effects exist. I would advise the authors to tone down their conclusions or add extra data to fully prove their points. Below

I highlight issues that need to be addressed prior to publication:

#1. Figure 4F. exogenous Kifc1 does not seem to get stabilised in response to DNA damage, although it gets phosphorylated. Can the authors explain why that is the case? A lot of the work they do is based on exogenously expressed kifc1.

Response: Thank you for your suggestion. In Fig. 4F of the original figure, the 293T cells transfected with Flag-KIFC1 were treated with etoposide for 4 h (Fig. 4 legends, p. 18 of the original manuscript). After etoposide treatment, the KIFC1-S26 phosphorylation was dramatically induced (about 4.8-8.9 fold enhancement) at 2-4 h (new Fig. 4J), whereas the KIFC1 protein levels showed a slight increase at 4 h (about 1.5 fold) and a marked increase at 8-15 h (about 2.8-4.8 fold). The following figure shows etoposide treatment-induced endogenous KIFC1 proteins accumulation for indicated times.

Figure legend. MDA-MB-231 cells were treated with etoposide (20 μ M) for the indicated times, and cell lysates were immunoblotted with antibodies against KIFC1 and β -actin (as an internal standard).

#2. Figure 4G. what about interaction with ATM/ATR upon DNA damage? As for the exogenous kifc1 that does not affect but for the endogenous thigs might be different.

Response: Thank you for your thoughtful suggestion. We performed new experiment using endogenous KIFC1 immunoprecipitation, and the result is shown in the new Fig. 4F. The data showed that the binding of endogenous KIFC1 to ATM and ATR upon DNA damage treatment was enhanced coupled with a increase of KIFC1 protein level, suggesting that etoposide treatment did not alterate the KIFC1-ATM/ATR binding affinity, which is similar to our original experiment result shown in the original manuscript (Fig. 4 F). To clearly show the endogenous KIFC1-ATM/ATR binding under etoposide treatment, we used the new Fig. 4F in the revised manuscript.

#3. Figure 4. Overall it would be helpful to add quantifications to the western blot data. For example 4H.

Response: Thank you for your suggestion. We have revised these figures, as shown in the

new Fig. 4G (Fig. 4H of the original figures), 4J, and 4K.

#4. The identification of the phospho site and phospho-kifc1 antibody data is very nice. However, it will be critical to demonstrate that this is also observed in endogenous kifc1. I don't think the authors show that. Given my point #1 above, it is important to know if this site is also phosphorylated in endogenous protein in response to DNA damage.

Response: Thank you for your suggestion. We used endogenous KIFC1 immunoprecipitation to examine KIFC1-S26 phosphorylation in the new Fig. 4J. The result showed that endogenous KIFC1-S26 phosphorylation was activated by etoposide treatment using the KIFC1^{S26p} antibody.

#5. Some controls are missing throughout this manuscript. It is essential that the authors show the percentage of centrosome amplification upon treatments, overexpressions etc. Not only the clustering versus multipolar phenotypes. Do the DNA damage treatments used here really induce bona fide centrosome amplification? what about cell lines that express the mutant S26A? in principle these would not cluster well and thus I would expect that the percentage of centrosome amplification would decrease overtime.

Response: Thank you for your suggestion. The percentage of centrosome amplification was the total percentage of pseudo-bipolar (clustering) and multipolar (non-efficient clustering) in the figures. We therefore showed the percentage of centrosome amplification (> 2 centrosomes, CA) using new graphs in the new Fig. S3A, Fig. 5D, and Fig. 7C. These results showed that the percentage of centrosome amplification (> 2 centrosomes per cell) increased after DNA damage treatment (36% of CA at 0 h, 41% of CA at 15 h, and 50% of CA at 48 h, shown in the new Fig. S3A), which is consistent with the results of previous researches [6-8]. The possible reason was that centrosome over-duplication occurred at G2 phase [6] and needed many centrosome-related proteins to assemble centrosomes.

The ratio of cells showing centrosome clustering to CA markedly increased at 15 h and 48 h after etoposide treatment (Fig. S3B). Etoposide treatment for 48 h always led to the production of apoptotic cells, in which there were chromosome agglutination and chaotic spindles. Thus, we prefer to treat cells with etoposide for 15 h, in order to precisely and conveniently examine the frequency of centrosome clustering.

The percentage of centrosome amplification in S26A stable cells (~25%) was decreased compared with wild-type (~35%) and S26D (~42%) stable cell lines in the original Fig. 5D (new Fig. 5F) (the total percentage of pseudo-bipolar and multipolar). We used a new graph to

clearly show the percentage of centrosome amplification in the new Fig. 5D, and found that the percentage of centrosome amplification was increased in all stable cell lines at 48 h after etoposide treatment.

#6. One of my main criticisms is the interpretation of the tumour data. there is no proof that the S26A mutants are more sensitive to etoposide due to inefficient clustering. We do not know the percentage of centrosome amplification in any of the conditions.

Response: Thank you for your suggestion. We showed that the percentage of centrosome amplification (CA) was increased after etoposide treatment at 48 h (in the new Fig. 5D). In the absence of etoposide, ~14% of KIFC1-S26A mitotic cells and ~12% of KIFC1-WT mitotic cells underwent cell death or cell cycle arrest via multipolar mitosis (Fig. 5E), because that the progeny of these cells are typically inviable [9]. In the presence of etoposide, the percentage of cells with multipolar mitosis in KIFC1-S26A mitotic cells was increased to ~26%, while that in KIFC1-WT mitotic cells was decreased to ~2% (as shown in the following table), indicating that etoposide treatment induced ~24% of mitotic cells avoiding multipolar mitosis and surviving via the activation of KIFC1-S26 phosphorylation. The result of Fig. 6A was in agreement with the abovementioned inference. Thus, we conclude that KIFC1-S26 phosphorylation leads to etoposide resistance in tumor cells via etoposide-induced centrosome amplification and centrosome clustering.

Stable cell lines	Untreatment			Etoposide treatment		
	WT	S26A	S26D	WT	S26A	S26D
% of mitotic cells with CA	36%	25%	42%	50%	42%	52%
% of multipolar mitosis	12%	14%	2%	2%	26%	2%
Cell death/ cell cycle arrest via multipolar mitosis	12%	14%	2%	2%	26%	2%

#7. I am not convinced by the conclusion that the combination of VE-822+etoposide has anything to do with centrosome amplification and clustering. Hasn't it been shown that DNA damage agents in combinations with agents that prevent DNA repair lead to better killing of tumour cells? There is no data to support that this combination is working via preventing centrosome clustering, even if clustering is inhibited. One way to assess that would be to do the combination therapy in the S26D mutant cells. But even in this case a better characterisation of this mutant in cells with normal centrosomes would be necessary.

Response: Thank you for your thoughtful suggestion. We agree with the reviewer's comment that VE-822 prevented DNA repair to promote cell death upon DNA damage. In this study,

we further used the combination of VE-822 and etoposide in KIFC1-rescue WT, S26A, and S26D cell lines. The result showed that VE-822 treatment dramatically increased etoposide-induced cell death in KIFC1-WT cells, but slightly increased etoposide-induced cell death in the KIFC1 mutants (S26A and S26D) cells (in the new Fig. 6B). Given VE-822 inhibited KIFC1-S26 phosphorylation in KIFC1-WT cells but not in KIFC1 mutants cells, we concluded that VE-822 sensitized etoposide treatment not only via preventing DNA repair but also via the inhibition of KIFC1-S26 phosphorylation.

In addition, we showed the characterization of KIFC1-WT and mutant cell lines as answered in question #8. Briefly, there were no significant changes in cellular location, spindle morphology, the average duration of mitosis, and lagging chromosomes during anaphase in these stable cells with normal centrosomes. There was also no significant change in the rate of cell proliferation of the KIFC1 (WT, S26A, and S26D)-rescued MCF-10A cell lines (the percentage of centrosome amplification was ~6 % – 9% in MFC-10A stable cell lines). Collectively, KIFC1-S26 phosphorylation plays a pivotal role in centrosome clustering in cells with amplified centrosomes but not in cells with normal centrosomes. Thus, we confirm that the inhibition of centrosome clustering by VE-822 treatment contributes to sensitization in DNA-damaging therapy. Notably, VE-822 killed these cells with amplified centrosomes, which were related to chromosomal instability and high risk of tumor recurrence.

#8. This brings me to another crucial point. It will be important to describe the role of these kifc1 mutants in cells with normal centrosomes. Kifc1 is dispensable to mitosis in cells with normal centrosomes but when overexpressed is leads to microtubule bundles in interphase. The authors should show images of interphase cells and a characterisation of mitosis in cells with normal centrosomes when these mutants are expressed.

Response: Thank you for your suggestion. When KIFC1 wide-type (WT) or mutant plasmids were transfected in tumor cells, we found that KIFC1 overexpression led to microtubule bundles and longer spindles (Fig. S6A) before the establishment of rescued KIFC1 WT and mutant stable cell lines. All overexpressions of GFP-KIFC1-WT, S26A, and S26D were mainly located in the nucleus in interphase and located in the centrosome and spindle in mitosis (Fig. S6B). They all induced microtubule bundles (~ 10 % – 16%) accompanied by condensed DNA, and most of them showed longer spindles during mitosis (Fig. S6B). In my opinion, redundant KIFC1, which was induced by overexpression, was the major reason for microtubule bundles and longer spindles.

To avoid the negative effects of KIFC1 overexpression, MDA-MB-231 and MCF-10A

cells were infected with low-titer lentivirus and then selected with puromycin and hygromycine, and finally the stable cell lines with exogenous KIFC1 protein levels similar to the endogenous KIFC1 in MDA-MB-231 or MCF-10A cells were selected for further experiments (in the Materials and Methods, Establishment of stable cell lines, p. 12 of the original manuscript). KIFC1 or KIFC1 mutants in KIFC1-WT, S26A, and S26D stable cell lines were localized to the centrosome, nucleus, and cytoplasm, in a similar manner as the localization of endogenous KIFC1 in cells with normal centrosomes. KIFC1 or KIFC1 mutants did not lead to microtubule bundles and longer spindles in these stable cell lines under normal culture conditions. Moreover, there were no significant changes in lagging chromosomes during anaphase and the average duration of mitosis in these stable cells with normal centrosomes. There was also no marked change in the rate of cell proliferation of the KIFC1 (WT, S26A, and S26D)-rescued MCF-10A cell lines. Thus, it is important to control the degree of KIFC1 expression for the study of KIFC1.

#9. Related with point #8 above, the CIN data in Figure & suggests that these mutants might be doing something else n mitosis leads to increase lagging of chromosomes. This is more evident in 7C where authors use MCF10A cells that have normal centrosome number. What is the actual % centrosome amplification upon etoposide treatment in these cells? And why aren't the authors quantifying these defects in WT, S26A and S26D in the absence of etoposide? This is crucial to know if the effects depend on drug treatment and centrosome amplification. Chromosome segregation should be quantifying in combination with centrin staining to assess whether its only visible in cells with extra centrosomes.

Response: Thank you for your thoughtful suggestion. We have added these data in the new Fig. 7. We found that the percentage of centrosome amplification in MCF-10A cells was ~8% and further increased to ~30% after etoposide treatment, which was consistent with the results of Pellman [8, 10]. The percentages of centrosome amplification, centrosome clustering, and aneuploidy in the absence or presence of etoposide were shown (new Fig. 7C, D, E, and F).

The percentage of centrosome amplification in WT, S26A, and S26D stable MCF10A cells was ~ 6.5% – 9%, and further increased to ~ 27% – 35% (in the new Fig. 7C). Etoposide treatment significantly increased the proportion of centrosome clustering to non-efficient clustering in cells with wild-type KIFC1 (ShN and KIFC1-WT) but not in cells with KIFC1 mutants (S26A and S26D) (in the new Fig. 7D). Increased centrosome amplification and clustering in KIFC1-WT and S26D cells induced lagging chromosomes and led to chromosomal instability (CIN).

Without etoposide treatment, the percentage of aneuploidy in KIFC1-WT and -S26A cells was low and similar to that in untreated MCF-10A cells, indicating that etoposide-induced centrosome amplification and clustering was necessary for the occurrence of CIN (Fig. 7E and F). Notably, the percentage of aneuploidy had no significant change in KIFC1-S26A cells with or without a low-concentration of etoposide treatment, and was markedly increased in KIFC1-WT and KIFC1-S26D cells after etoposide treatment, indicating that etoposide-induced centrosome amplification and clustering played a pivotal role in the occurrence of aneuploidy.

In the analysis of the chromosomal number, tumor cells were initially swollen in the KCl solution before being fixed in Carnoy's solution. The cells with dispersed chromosomes were then dropped onto slides, dried, and stained with Giemsa. For the staining of centrin, the cells were fixed with cold methanol, permeabilized, and incubated with primary antibodies and secondary antibodies. We failed to stain the centrin and chromosomes on the same glass slides, and the dispersed chromosomes were easily detached from the slides after incubation with primary and secondary antibodies. In this study, we have found that etoposide treatment induced a significantly increased frequency of aneuploidy in KIFC1-WT and S26D cells but not in KIFC1-S26A cells (Fig. 7E), and Fig. 5 and Fig. 7 show the crucial function of KIFC1-S26 phosphorylation in cells with amplified centrosomes but not in the cells with normal centrosomes. Given the high frequency of lagging chromosomes in tumor cells with extra centrosomes [9], we concluded that centrosome clustering in cells with amplified centrosomes induced by etoposide treatment was a critical reason of the increased frequency of CIN.

#10. Figure 8E it is impossible to see all the treatments in the graph.

Response: Thank you for your suggestion. We have revised the graph in Fig. 8E to show all treatments.

#11. Figure 8. Did the authors assess the toxicity of CW069? They mention that this inhibitor causes unspecific toxicity, as it has been reported. Thus, the use of this inhibitor here is confusing. I am also intrigued to why if clustering is prevented by VE-822 and CW069 more aneuploidy is not observed? Preventing clustering does not immediately kill cells, it takes in fact several days, so not sure I understand this.

Response: Thank you for your suggestion.

First, we have confirmed the nonspecific toxicity of CW069, which was computationally designed based on inhibitors of kinesin Eg5 [1]. Although CW069 or VE-822 was not a

specific inhibitor of KIFC1 or KIFC1-S26 phosphorylation, they both prevented tumor recurrence through KIFC1 inhibition. We confirmed the significance of inhibition of KIFC1 and S26 phosphorylation using CW069 and VE-822; therefore, developing small molecule compounds to specifically target KIFC1-S26 phosphorylation is a promising therapeutic strategy to address tumor recurrence.

Second, we agree with the reviewer's comment that the cells with multipolar mitosis show high levels of aneuploidy when centrosome clustering is inhibited by VE-822 or CW069. Moreover, these cells have a prolonged mitosis and may survive for several days. However, the progeny of these cells is typically inviable, and 70 – 80% of these cells go to cell death [9]. With the proliferation of cancer cells, the ratio of survival cells with multipolar mitosis decreased. Thus, previous studies demonstrated that aneuploidy is induced by centrosome clustering but not multipolar mitosis in cancer cells with amplified centrosomes [9].

#. On page 2: "... either die to chromosome segregation...". This is not correct. As the authors mentioned, centrosome clustering allows survival while leading to chromosome missegregation. What the authors should say instead that high levels of aneuploidy generated by multipolar mitosis is the cause of cell death.

Response: Thank you for your thoughtful comments. We have made revisions as suggested, “Cancer cells with supernumerary centrosomes can either die due to high levels of aneuploidy generated by multipolar mitosis or divide and produce viable progeny by achieving a pseudo-bipolar structure via clustering their centrosomes into two functional poles, a process called centrosome clustering.” (Introduction, page 2)

#. On page 2: what do the authors mean by "exhibit a period of division"? this is very unclear to me.

Response: Thank you for the important question. The text has been revised to “Cancer cells undergoing centrosome clustering have a prolonged time to form a pseudo-bipolar spindle in which single kinetochores often attach to microtubules emanating from different poles, resulting in lagging chromosomes during anaphase [9].” (Introduction, page 2)

Once again, we thank you for the time you put in reviewing our paper and look forward to meeting your expectations.

References

1. Watts, C.A., et al., *Design, synthesis, and biological evaluation of an allosteric inhibitor of HSET that targets cancer cells with supernumerary centrosomes*. Chem

- Biol, 2013. **20**(11): p. 1399-410.
2. Wu, J., et al., *Discovery and mechanistic study of a small molecule inhibitor for motor protein KIFCI*. ACS Chem Biol, 2013. **8**(10): p. 2201-8.
 3. Kwon, M., et al., *Mechanisms to suppress multipolar divisions in cancer cells with extra centrosomes*. Genes & Development, 2008. **22**(16): p. 2189-2203.
 4. Shiloh, Y., *ATM and ATR: networking cellular responses to DNA damage*. Curr Opin Genet Dev, 2001. **11**(1): p. 71-7.
 5. Montecucco, A. and G. Biamonti, *Cellular response to etoposide treatment*. Cancer Lett, 2007. **252**(1): p. 9-18.
 6. Dodson, H., et al., *Centrosome amplification induced by DNA damage occurs during a prolonged G2 phase and involves ATM*. EMBO J, 2004. **23**(19): p. 3864-73.
 7. Bourke, E., et al., *DNA damage induces Chk1-dependent centrosome amplification*. EMBO Rep, 2007. **8**(6): p. 603-9.
 8. Loffler, H., et al., *DNA damage-induced centrosome amplification occurs via excessive formation of centriolar satellites*. Oncogene, 2013. **32**(24): p. 2963-72.
 9. Ganem, N.J., S.A. Godinho, and D. Pellman, *A mechanism linking extra centrosomes to chromosomal instability*. Nature, 2009. **460**(7252): p. 278-282.
 10. Godinho, S.A., et al., *Oncogene-like induction of cellular invasion from centrosome amplification*. Nature, 2014. **510**(7503): p. 167-171.

Reviewers' Comments:

Reviewer #1:

Remarks to the Author:

Thank you addressing the comments I raised and for the extra work you put in to addressing them. I would be glad to accept this manuscript as long as you clearly stipulate the doses of AZD1390 and AZD6738 used. 5uM is a very high dose for AZD1390 as this is an extremely potent inhibitor of ATM with a cellular IC50 of about 3nM. You are very likely to introduce off-target effects at 5uM, if that is the concentration you used, because other kinases are hit >1uM. Please include data below 1uM otherwise you have not addressed the risks of using other less selective ATM inhibitors used in your previous version. If you still see the same effects at lower doses and which disappear at very low doses (<1uM) then I will accept the effect is dose-dependent and ATM-specific, to mimic the shATM effects. Sometimes pharmacological inhibition of an enzyme does not phenocopy genetic or protein silencing and if the argument is to use ATM or ATR inhibitors to target centrosome de-clustering, then the correct doses should be used. Thank you.

Reviewer #2:

Remarks to the Author:

Overall, the authors did a great job addressing the comments and concerns raised. They added extra experimental data and new quantifications that make the story stronger. There are only few of the original points that I don't think have been sufficiently addressed/explained.

Related to #9. There seems to be a mis characterization of the work of Ganem et al, that the authors reference here. This is notorious in several parts of this manuscript and I would urge the authors to correct this. It has been well established by that paper and by others in the field that multipolar anaphases have massively increased chromosome missegregation when compared with clustered anaphases. This is a fact and I do not think it is up to interpretation. Thus, when the authors show on figure 7A that anaphases from KIFC1 S26A mutant have less missegregated chromosomes that raises a red flag. Which anaphases are the authors looking at? I am assuming anaphases from cells with extra centrosomes to be comparable? So how come there is less missegregation? What has been proposed is that missegregation of chromosomes result from the merotelic attachments generated during multipolar metaphases. Thus, in multipolar anaphases those are more apparent. However, even in clustered anaphases there is a low rate of chromosome missegregation leading to aneuploidy. What has been shown by several labs is that multipolar divisions are not viable and therefore it is assumed that chromosome missegregation generated during clustered mitosis will be contributing to the viable aneuploid population. This is different from saying that multipolar mitosis do not generate chromosome missegregation and aneuploidy. This is incorrect. Again, in figures 7E and 7F the authors show the opposite, that in the S26A mutant where multipolar mitoses are increased there is less aneuploidy. This would be true only in long term experiments where multipolar divisions would lead to cell death... but not in a 15 hr treatment with etoposide.

Related to #9 and #6. In several graphs, there seems to be a significant lower number of metaphase cells with amplified centrosomes in the S26A mutant, even after etoposide short term treatment. I am wondering if there is another parallel issue going on in this mutant. For example, whether cells with extra centrosomes are arresting more in S26A mutant? Do the S26A mutant cells have increased DNA damage? And this could impact the in vivo data as well? And could be responsible for the increase sensitivity to etoposide? Not only failure to cluster? This has been a concern that I raised before. There is only circumstantial evidence to suggest that the effects observed in vivo are only due to cluster efficiency. Note that the treatments are performed for long period of times in vivo than in vitro.

Related to #7. I maintain that at this stage the authors cannot conclude that the effects observed

in vivo due to the combination of VE-822 and etoposide cannot be ascribed to clustering efficiency. To prove this, the authors will need to show that in vivo tumours containing extra centrosomes are more sensitive to VE-822 than tumours with normal centrosomes number, independently of etoposide treatment. I agree it is an interesting possibility but not fully proven.

Sep 19, 2020

Re: Revision to *Nature Communications*

Dear Reviewers:

We deeply appreciate the time and effort you have spent in reviewing our manuscript. In the new manuscript, we supplemented 3 new figures (Supplemental Fig. S7, S8 and S9). The following is a point-by-point response to the questions and comments.

Responses to critiques (reviewers' comments are italic and underlined):

Reviewer #1 (Remarks to the Author):

Thank you addressing the comments I raised and for the extra work you put in to addressing them. I would be glad to accept this manuscript as long as you clearly stipulate the doses of AZD1390 and AZD6738 used. 5uM is a very high dose for AZD1390 as this is an extremely potent inhibitor of ATM with a cellular IC50 of about 3nM. You are very likely to introduce off-target effects at 5uM, if that is the concentration you used, because other kinases are hit > 1uM. Please include data below 1uM otherwise you have not addressed the risks of using other less selective ATM inhibitors used in your previous version. If you still see the same effects at lower doses and which disappear at very low doses (<1uM) then I will accept the effect is dose-dependent and ATM-specific, to mimic the shATM effects. Sometimes pharmacological inhibition of an enzyme does not phenocopy genetic or protein silencing and if the argument is to use ATM or ATR inhibitors to target centrosome de-clustering, then the correct doses should be used. Thank you.

Response: Thank you for your suggestion regarding the correct dosage of AZD1390. In this study, AZD1390 was used at 20 nM to inhibit ATM. As indicated in the legend of Fig. 4D, page 19 of the old manuscript and page 21 of the new manuscript, the cells were treated with AZD1390 (20 nM), VE-822 (5 μ M), and AZD6738 (25 nM). Thus, there may be a misunderstanding regarding this issue. Because VE-822 inhibits ATM and ATR with IC₅₀ at 2.6 μ M and 19 nM respectively [1], we used VE-822 at 5 μ M to inhibit ATM and ATR. We would like to thank the referee again for taking the time to review our manuscript.

Reviewer #2 (Remarks to the Author): Overall, the authors did a great job addressing the comments and concerns raised. They added extra experimental data and new quantifications that make the story stronger. There are only few of the original points that I don't think have been sufficiently ddressed/explained.

Question 1: Related to #9. There seems to be a mis characterization of the work of Ganem et al, that the authors reference here. This is notorious in several parts of this manuscript and I would urge the authors to correct this. Is has been well established by that paper and by

others in the field that multipolar anaphases have massively increased chromosome missegregation when compared with clustered anaphases. This is a fact and I do not think it is up to interpretation. Thus, when the authors show on figure 7A that anaphases from KIFC1 S26A mutant have less missegregated chromosomes that raises a red flag. Which anaphases are the authors looking at? I am assuming anaphases from cells with extra centrosomes to be comparable? So how come there is less missegregation? What has been proposed is that missegregation of chromosomes result from the merotelic attachments generated during multipolar metaphases. Thus, in multipolar anaphases those are more apparent. However, even in clustered anaphases there is a low rate of chromosome missegregation leading to aneuploidy. What has been show by several labs is that multipolar divisions are not viable and therefore it is assumed that chromosome missegregation generated during clustered mitosis will be contributing to the viable aneuploid population. This is different from saying that multipolar mitosis do not generate chromosome missegregation and aneuploidy. This is incorrect. Again, in figures 7E and 7F the authors show the opposite, that in the S26A mutant where multipolar mitoses are increased there is less aneuploidy. This would be true only in long term experiments where multipolar divisions would lead to cell death... but not in a 15 hr treatment with etoposide.

Response: Thank you for your thoughtful suggestion and we agree with reviewer's comment. The work of Ganem et al. and many published articles have shown that multipolar anaphase leads to higher levels of chromosome missegregation than centrosome clustering in cancer cells with supernumerary centrosomes [2, 3]. Because the cells with multipolar mitosis may die due to high levels of aneuploidy, and the progeny of these cells are typically not viable. In contrast, tumor cells undergoing centrosome clustering can survive and further proliferate. Lagging chromosomes induced by centrosome clustering is meaningful for the occurrence of aneuploidy in the growing tumors. Thus, we just assessed the percentage of lagging chromosomes induced by centrosome clustering in Fig. 7A. Representative images showed bipolar or pseudo-bipolar mitosis (bipolar spindles) and excluded multipolar mitosis (multipolar spindles) in Fig. 7A. Centrosome clustering led to a significantly increased frequency of lagging chromosomes (~42%) during anaphase in cells with extra centrosomes [3]. In the new manuscript and figure legends, we modified the text as following (the modification is underlined).

1. Page 9, line 2 to 16: Multipolar mitosis leads to higher levels of aneuploidy than pseudo-bipolar mitosis (centrosome clustering) in tumor cells with supernumerary centrosomes. However, the cells with multipolar mitosis may die due to high levels of aneuploidy, and the progeny of these cells are typically not viable [3]. In contrast, tumor cells undergoing centrosome clustering can survive and further proliferate. With the proliferation of cancer cells, lagging chromosomes induced by centrosome clustering promote CIN [3-6].

Thus, we next assessed the effect of KIFC1-S26 phosphorylation on the percentage of lagging chromosomes induced by centrosome clustering. Immunofluorescence staining analysis showed that the percentage of lagging chromosomes induced by centrosome clustering in the KIFC1-S26D cell line (~25%) was significantly higher than that in the KIFC1-S26A cell line (~8%). Etoposide treatment led to an increased frequency of lagging chromosomes induced by centrosome clustering in KIFC1-WT and normal control cells, but not in KIFC1 mutants (S26A and S26D) cells. VE-822 treatment markedly decreased etoposide-induced enhancement of the frequency of lagging chromosomes in KIFC1-WT and normal control cells (Fig. 7A). These results indicate that KIFC1-S26 phosphorylation leads to a high tendency for lagging chromosomes induced by centrosome clustering.

2. Page 22, line 27 to 32: (A) Representative images showing lagging chromosomes induced by centrosome clustering in the MDA-MB-231 stable cell lines. The cells were pretreated with VE-822 for 1 h, and then treated with etoposide (2 μ M) for another 15 h. Spindle and DNA were co-stained with DAPI and α -tubulin. In quantitative analyses, anaphase cells with lagging chromosomes induced by centrosome clustering were compared with all anaphase cells. More than 100 cells per experimental group were counted. Scale bar, 10 μ m. * $P < 0.05$; ** $P < 0.01$.

Question 2: Related to #9 and #6. In several graphs, there seems to be a significant lower number of metaphase cells with amplified centrosomes in the S26A mutant, even after etoposide short term treatment. I am wondering if there is another parallel issue going on in this mutant. For example, whether cells with extra centrosomes are arresting more in S26A mutant? Do the S26A mutant cells have increased DNA damage? And this could impact the in vivo data as well? And could be responsible for the increase sensitivity to etoposide? Not only failure to cluster? This has been a concern that I raised before. There is only circumstantial evidence to suggest that the effects observed in vivo are only due to cluster efficiency. Note that the treatments are performed for long period of times in vivo than in vitro.

Response: Thank you for your thoughtful suggestion. To test the possibility of another parallel issues, we analyzed cell cycle progression [7], aberrant nuclear membrane [7], DNA damage, and an acentrosomal spindle organization [8] in KIFC1-rescued WT, S26A, and S26D cell lines. As shown in Fig. S7A, KIFC1-S26A cells showed slightly prolonged S and G2/M phases compared with KIFC1-WT and -S26D cells (Fig. S7A). The prolonged S phase may suggest that KIFC1 is associated with nuclear importins and transported recombinant/replicate-related proteins during S phase [7, 9]. There was no significant difference in the percentage of aberrant nuclear membranes in KIFC1-rescued stable cell lines with normal 2

centrosomes or > 2 centrosomes (Fig. S7B). As shown in Fig. S7C, γ H2AX staining (a DNA damage marker [10]) indicated that KIFC1-WT, S26A, and S26D mutant cells did not show serious DNA damage except some dying cells under normal culture conditions. Nevertheless, the slightly prolonged S phases and aberrant nuclear membrane were not responsible for the increase sensitivity to etoposide.

We further analyzed acentrosomal spindle organization in KIFC1-rescued MDA-MB-231 cell lines. Acentrosomal spindle organization is formed by centrosome-independent mechanisms, and KIFC1 is a key driver to crosslink and focus MT-minus ends into two spindle poles [8, 11]. The majority of cancer cells containing two centrosomes associated with normal centrosomal poles (Fig. S8A, illustrated by gray spindle and bars), and a small fraction of cancer cells (~4%) formed bipolar spindles with acentrosomal poles (illustrated by green spindle and bars, free centrosome). The percentage of acentrosomal poles in cells with two centrosomes was significantly increased to ~30% after DNA damage for 48 h (Fig. S8A), in accordance with the work of Kleylein-Sohn et al. [8]. However, we found that KIFC1-S26A cells with two centrosomes showed a significant increase in the frequency of multipolar spindles containing additional pole structures devoid of bona fide centrosomes (illustrated by red spindle and bars) after DNA damage (Fig. S8A). In our study, to elucidate the sensitivity of etoposide in KIFC1-S26A cells, we just assessed acentrosomal poles in cells with two centrosomes although acentrosomal poles were also found in cells with extra centrosomes. In addition, the concentration of etoposide in the experiment of etoposide-induced cell death did not decrease the ratio of G2/M phase (Fig. S8B), which was the sensitive phase to etoposide treatment [12]. Centrosome clustering and acentrosomal spindle organization also occurred in M phase. Collectively, we concluded the following: (1) DNA damage led to centrosome amplification and centrosome clustering, and the cells with supernumerary centrosomes (~40%) in KIFC1-S26A cells were sensitive to etoposide treatment through non-efficient centrosome clustering; and (2) DNA damage led to acentrosomal spindle organization, and the cells with normal two centrosomes (~60%) in KIFC1-S26A cells were sensitive to etoposide treatment through acentrosomal poles-induced multipolar spindles formation.

Accordingly, increased apoptosis (the activated caspase-3 and the cleaved PARP) was detected in KIFC1-S26A cells (Fig. S8C). Xenograft tumors underwent a longer period of etoposide treatment, and we found that more tumor cells with two centrosomes showed acentrosomal poles after DNA damage in our *in vivo* experiment (Fig. S8D). A long period of etoposide treatment *in vivo* resulted in tumor cells undergoing more cell cycle progression, and was advantageous to centrosome clustering in cells with extra centrosomes and

acentrosomal poles in cells with two centrosomes. Thus, we concluded that non-efficient centrosome clustering (in cells with extra centrosomes) and acentrosomal poles-induced multipolar spindles (in cells with two centrosomes) in KIFC1-S26A cells were responsible for increased sensitivity to etoposide *in vivo* and *in vitro*.

Question 3: *Related to #7. I maintain that at this stage the authors cannot conclude that the effects observed in vivo due to the combination of VE-822 and etoposide cannot be ascribed to clustering efficiency. To prove this, the authors will need to show that in vivo tumours containing extra centrosomes are more sensitive to VE-822 than tumours with normal centrosomes number, independently of etoposide treatment. I agree it is an interesting possibility but not fully proven.*

Response: Thank you for your excellent suggestion. To further study the inhibitory function of VE-822 on tumor growth, we used a HeLa monoclonal cell line with inducible centrosome amplification (CA) (TETON-PLK4 cell line, the percentage of CA was from 11% up to ~80% after doxycycline treatment) [5, 6, 13]. PLK4 overexpression and centrosome amplification were markedly induced in the cell line after doxycycline treatment (Fig. S9A, B). Xenograft tumors from the cells with inducible CA were treated with PBS, VE-822, or doxycycline for 6 weeks. As shown in Fig. S9C, VE-822 dramatically inhibited the volumes and weights of tumors in presence of doxycycline (DOX), but not in absence of doxycycline. There is no significant decrease in the volumes and weights of tumors from normal HeLa cells (the percentage of CA was ~11%) after VE-822 and DOX treatment (Fig. S9D). In addition, the growth of xenograft tumors from MDA-MB-231 cells (with high-frequency of CA) was inhibited by VE-822 treatment (Fig. S9E). Collectively, these new supplementary data suggest that VE-822 treatment inhibits the growth of tumors with high-frequency of CA.

References

1. Fokas, E., et al., *Targeting ATR in vivo using the novel inhibitor VE-822 results in selective sensitization of pancreatic tumors to radiation*. Cell Death Dis, 2012. **3**: p. e441.
2. McGranahan, N., et al., *Cancer chromosomal instability: therapeutic and diagnostic challenges*. EMBO Rep, 2012. **13**(6): p. 528-38.
3. Ganem, N.J., S.A. Godinho, and D. Pellman, *A mechanism linking extra centrosomes to chromosomal instability*. Nature, 2009. **460**(7252): p. 278-282.
4. Giam, M. and G. Rancati, *Aneuploidy and chromosomal instability in cancer: a jackpot to chaos*. Cell Division, 2015. **10**(1).
5. Kwon, M., *Using Cell Culture Models of Centrosome Amplification to Study Centrosome Clustering in Cancer*. 2016. **1413**: p. 367-392.
6. Godinho, S.A., et al., *Oncogene-like induction of cellular invasion from centrosome*

- amplification*. Nature, 2014. **510**(7503): p. 167-171.
7. Wei, Y.L. and W.X. Yang, *Kinesin-14 motor protein KIFC1 participates in DNA synthesis and chromatin maintenance*. Cell Death Dis, 2019. **10**(6): p. 402.
 8. Kleylein-Sohn, J., et al., *Acentrosomal spindle organization renders cancer cells dependent on the kinesin HSET*. J Cell Sci, 2012. **125**(Pt 22): p. 5391-402.
 9. Farina, F., et al., *Kinesin KIFC1 actively transports bare double-stranded DNA*. Nucleic Acids Res, 2013. **41**(9): p. 4926-37.
 10. Shiloh, Y. and Y. Ziv, *The ATM protein kinase: regulating the cellular response to genotoxic stress, and more*. Nat Rev Mol Cell Biol, 2013. **14**(4): p. 197-210.
 11. Verhey, K.J. and J.W. Hammond, *Traffic control: regulation of kinesin motors*. Nat Rev Mol Cell Biol, 2009. **10**(11): p. 765-77.
 12. Luzhin, A.V., et al., *Automated Analysis of Cell Cycle Phase-Specific DNA Damage Reveals Phase-Specific Differences in Cell Sensitivity to Etoposide*. J Cell Biochem, 2016. **117**(10): p. 2209-14.
 13. Raab, M.S., et al., *GF-15, a Novel Inhibitor of Centrosomal Clustering, Suppresses Tumor Cell Growth In Vitro and In Vivo*. Cancer Research, 2012. **72**(20): p. 5374-5385.

Reviewers' Comments:

Reviewer #1:

Remarks to the Author:

Thank you for clarifying the ATM and ATR inhibitor dosing concentrations. The extra supplemental data you have now included also adds strength to the dataset. I am happy to approve this manuscript.

Reviewer #2:

Remarks to the Author:

I would like to thank the authors for the extra effort to address my comments. In my opinion the manuscript has been greatly improved.

In particular the new in vivo experiment with the PLK4-induced extra centrosomes is very important to demonstrate that indeed tumours with extra centrosomes are more sensitive to VE-822.

I have no other comments and i believe this work should be published in its current form.